# Fibre-specific mitochondrial protein abundance is linked to resting and post-training mitochondrial content in the muscle of men

Elizabeth G. Reisman [1,2,13], Javier Botella [1,3,13], Cheng Huang[4], Ralf B. Schittenhelm [4], David A. Stroud [5,6,7], Cesare Granata [1,8,9,10], Owala S. Chandrasiri[1], Georg Ramm [11], Viola Oorschot[11,12], Nikeisha J. Caruana [1,5] ✉ & David J. Bishop [1] ✉

Analyses of mitochondrial adaptations in human skeletal muscle have mostly used whole-muscle samples, where results may be confounded by the presence of a mixture of type I and II muscle fibres. Using our adapted mass spectrometry-based proteomics workflow, we provide insights into fibre-specific mitochondrial differences in the human skeletal muscle of men before and after training. Our findings challenge previous conclusions regarding the extent of fibre-type-specific remodelling of the mitochondrial proteome and suggest that most baseline differences in mitochondrial protein abundances between fibre types reported by us, and others, might be due to differences in total mitochondrial content or a consequence of adaptations to habitual physical activity (or inactivity). Most training-induced changes in different mitochondrial functional groups, in both fibre types, were no longer significant in our study when normalised to changes in markers of mitochondrial content.

The remarkable ability of skeletal muscle to adapt to repeated contractile stimuli is one of the most fundamental and intriguing aspects of physiology[1]. Repeated exercise sessions (i.e. exercise training) alter the expression of proteins, which can lead to improved maintenance of cellular homoeostasis[2], altered metabolism[3] and the prevention and treatment of many chronic diseases[4]. Exercise is also a potent stimulus to activate mitochondrial biogenesis[5–7], with resulting increases in mitochondrial protein content and respiratory function[8,9]. These

[1]Institute for Health and Sport (IHES), Victoria University, Melbourne, VIC, Australia. [2]Mary MacKillop Institute for Health Research, Australian Catholic University, Melbourne, VIC, Australia. [3]Metabolic Research Unit, School of Medicine and Institute for Mental and Physical Health and Clinical Translation (IMPACT), Deakin University, Waurn Ponds, VIC, Australia. [4]Monash Proteomics & Metabolomics Facility, Biomedicine Discovery Institute and Department of Biochemistry and Molecular Biology, Monash University, Clayton, VIC, Australia. [5]Department of Biochemistry and Pharmacology, Bio21 Molecular Science and Biotechnology Institute, The University of Melbourne, Parkville, VIC, Australia. [6]Murdoch Children's Research Institute, Royal Children's Hospital, Parkville, VIC, Australia. [7]Victorian Clinical Genetics Services, Royal Children's Hospital, Parkville, VIC, Australia. [8]Department of Diabetes, Central Clinical School, Monash University, Melbourne, VIC, Australia. [9]Institute for Clinical Diabetology, German, Diabetes Center, Leibniz Center for Diabetes Research at Heinrich-Heine-University, Düsseldorf, Düsseldorf, Germany. [10]German Center for Diabetes Research, Partner Düsseldorf, München-Neuherberg, Germany. [11]Ramaciotti Centre for Cryo EM, Biomedicine Discovery Institute and Department of Biochemistry and Molecular Biology, Monash University, Clayton, VIC, Australia. [12]Electron Microscopy Core Facility, European Molecular Biology Laboratory, Heidelberg, Germany. [13]These authors contributed equally: Elizabeth G. Reisman, Javier Botella. ✉e-mail: nikeisha.caruana@unimelb.edu.au; David.Bishop@vu.edu.au

adaptations appear to depend on the characteristics of the exercise stimulus[10,11].

One limitation of most human training studies to date is the analysis of whole-muscle samples, where the results may be confounded by divergent changes in different fibre types[12]. Human skeletal muscle is composed of three major fibre types (type I, IIa, and IIx), which differ in their contractile, metabolic and bioenergetic characteristics; this is reflected by a greater reliance on mitochondrial oxidative enzyme activity in type I fibres[13–15] and greater glycolytic enzyme activity in type II fibres[16]. Furthermore, Henneman's size principle[17] proposes that type I fibres have a lower threshold of activation, and will be recruited at lower exercise intensities, whereas fast-twitch fibres (type IIa and IIx) have a higher activation threshold and will be increasingly recruited at greater exercise intensities[18,19]. Skeletal muscle fibre recruitment is therefore related to exercise intensity[18,19], and it has been hypothesised that higher exercise intensities might stimulate greater adaptations in type II fibres relative to lower intensities[17,20,21]. It has also been proposed that recruitment of type II fibres may help to explain the greater mitochondrial adaptations reported following high-intensity training[11,18,19,22]. Until recently, however, technical limitations have prevented the direct assessment of fibre-type-specific adaptations to different types of exercise training.

Modifications to immunoblotting protocols have enabled researchers to identify training-induced adaptations of mitochondrial proteins in single skeletal muscle fibres[23–26]. However, due to limited starting material, only five of the more than 1100 known mitochondrial proteins have been investigated to date[12,23–27]. Improvements in mass-spectrometry-based proteomic techniques have allowed greater resolution of the contractile and metabolic features of single skeletal muscle fibre types—first in mice[28] and then in humans[29]. The detection of functionally important, low-abundance proteins, in addition to thousands of other proteins, was a significant development that has advanced our understanding of the breadth and complexity of proteins at the fibre level in skeletal muscle. Recent work has built on this research and adapted this technique to investigate fibre-type-specific protein changes to moderate-intensity training in the skeletal muscle of a small number of men ($n = 5$)[30]. To date, however, no proteomic study has investigated the effects of divergent exercise intensities, which differ in their recruitment of the major fibre types, on fibre-type-specific mitochondrial adaptations. A greater understanding of fibre-type-specific adaptations to different exercise intensities may help to optimise the prescription of exercise to prevent and treat diseases, as well as counteract some of the detrimental effects of ageing.

In the present study, we developed a sensitive mass spectrometry-based (MS) proteomics workflow that allowed for the pooling of single muscle fibres, and the utilisation of tandem mass tag (TMT) labelling, to facilitate the identification of low abundant mitochondrial proteins. To investigate potential fibre-type-specific changes in protein abundances following exercise training, we employed two very different training interventions - moderate-intensity continuous training (MICT) and sprint-interval training (SIT). These two types of training were chosen as they differ in intensity (i.e. moderate versus very high intensity) and have been reported to require different skeletal muscle fibre recruitment patterns[22,27,31]. In contrast to the only published single-fibre proteomic study to examine the effects of exercise training[30], we did not detect any significant changes in the abundance of individual proteins in either fibre type following either training type. This finding, based on our more robust statistical approach, raises important questions regarding the ability of current single-fibre studies to avoid false positives and to confidently detect training-induced changes of individual proteins in single-fibre pools. While we were able to detect fibre-specific changes in different mitochondrial functional groups following training, many of these differences were no longer apparent when corrected for training-induced changes in mitochondrial content. Thus, most training-induced changes in mitochondrial protein abundances were no longer significant when normalised to the overall increase in mitochondrial content. Despite the hypothesis that SIT would target type II fibres, we observed few differentially expressed protein functional groups following SIT. There was, however, a decreased abundance of complex IV subunits in both type I and II fibres following SIT.

## Results

### Single-fibre proteomics

Only two studies in young men, both with small sample sizes ($n = 4$ and 5, respectively), have employed proteomics to describe fibre-type-specific differences for the contractile and metabolic features of human skeletal muscle fibres[29,30]. In the present study, we present the data from sixteen recreationally active, healthy men who completed one of two 8-week training interventions - either moderate-intensity continuous training (MICT; $n = 8$) or very high-intensity sprint-interval training (SIT; $n = 8$) (see Supplementary Data 1-Tab 1 for participant characteristics). Before (PRE) and after (POST) training, skeletal muscle biopsies were collected from the vastus lateralis muscle and single fibres were isolated, fibre-typed, pooled and analysed utilising the proteomic workflow described in Fig. 1a.

One of the challenges of analysing skeletal muscle is the large dynamic range of protein concentrations caused by the highly abundant sarcomeric proteins, which account for over 50% of the total protein content[28,32–34], while proteins from all other muscle compartments, including the mitochondria, are confined to the lower half of the abundance range[34]. Compared to typical whole-tissue proteomic studies, single-muscle fibres provide very low protein yields[28,35], which further complicates the identification of lower abundant proteins[34,36]. While previous studies have used label-free quantitative shotgun approaches after filter-aided sample preparation[29,30], we chose a workflow incorporating TMT labelling, which not only allowed us to increase sample size and throughput to accommodate our complex experimental design investigating two different exercise intensities but also exhibits higher quantitative precision than label-free quantification[37,38].

Our approach identified a total of 3141 proteins across all pooled samples, with 81% (2534 proteins) of these identified with high-confidence (false discovery rate [FDR] < 0.05) and >1 unique peptide (Supplementary Data 1-Tab 2). On average, we were able to reliably identify $1849 \pm 59$ (1477–1999) high-confidence proteins per pooled single-fibre sample (Fig. S1a, b, c, d, e; Supplementary Data 9-Tab 1). Proteins were removed from our dataset, as well as in comparative datasets from ref. 29 to ref. 30, if they contained more than 30% missing values, leaving a total dataset across all samples in our study of 1600 proteins (Supplementary Data 1-Tab 3). This cut-off is designed to remove the bulk of missing data, which can cause further downstream bias, but still allow appropriate imputation of missing values for analysis.

To further understand the characteristics of the fibre-specific proteome, we examined the cellular localisation of the proteins identified in each individual sample of pooled type I or type II fibres based on Gene Ontology Cellular Components. Gene Ontology (GO) enrichment analysis of our dataset (Fig. 1b, Supplementary Data 1-Tab 4) demonstrates good coverage of the structural and metabolic features of the skeletal muscle fibre proteome, including contractile, sarcomeric, ribosomal and mitochondrial proteins.

To assess the comprehensiveness of our protein detection in pooled single fibres, we compared our dataset to the two previous studies investigating the single muscle fibre proteome in humans (Fig. 1c, Supplementary Data 1-Tab 5)[29,30]. There were 844 proteins in common between the three studies, which may represent proteins that can more confidently be used to compare fibre types in human skeletal muscle. One of the most significantly enriched cellular components for

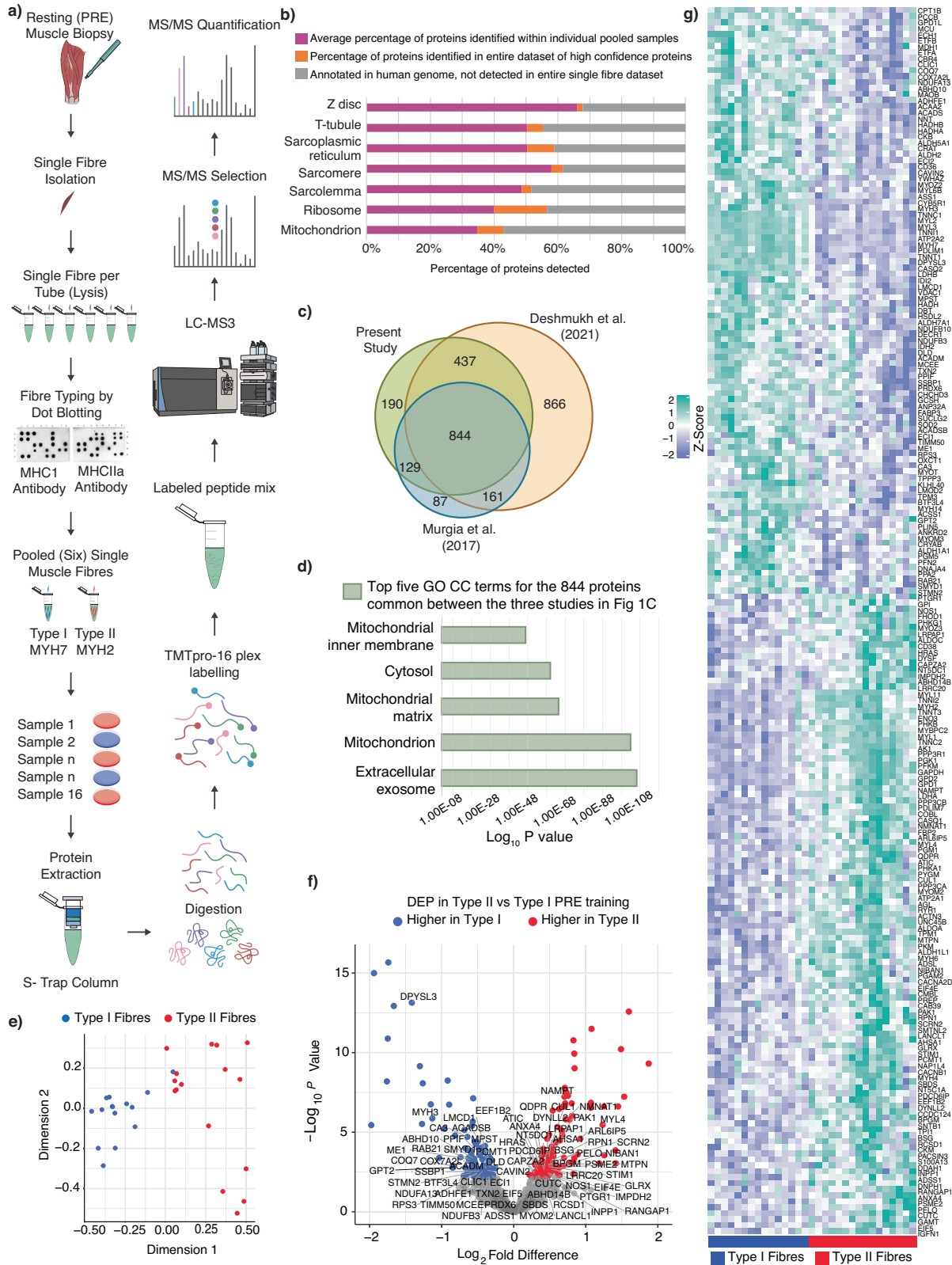

these commonly identified proteins was the mitochondrion (Fig. 1d, Supplementary Data 1-Tab 6).

## Fibre type differences in the pre-training samples

Multi-dimensional scaling revealed a good separation of the protein profiles between pooled type I and type II single fibre samples pre-training (Fig. 1e). Proteomic features of the different fibre-type pools

are presented in a volcano plot (Fig. 1f). A total of 206 proteins were identified as differentially expressed between fibre types pre-training (PRE), using an empirical Bayes method and with statistical significance set at an adjusted $P$ value cut-off of 0.05 (Benjamini-Hochburg correction) (Supplementary Data 1-Tab 7). Proteins were coloured if they had an adjusted $P < 0.05$ and a $\log_2$ Fold Difference of $>0.2$ (red, higher in Type II fibres) or $<-0.2$ (blue, higher in Type I

**Fig. 1 | Isolation of muscle fibres for proteomic analysis reveals fibre-type differences in the pre-training samples. a** Workflow for characterisation of pooled human skeletal muscle fibres. Single fibres obtained from human biopsy muscle samples and fibre typing confirmed through dot blotting. Six fibres were pooled, with proteins extracted followed by labelling with TMT. LC-MS3 performed to quantify proteins. **b** Percentage of proteins within samples in comparison with human genome. Bars display percentage of proteins identified in GOCC group based on (i) average number of proteins within individual pooled samples (pink), (ii) number of proteins within entire dataset of high-confidence proteins (orange) and (iii) known annotated proteins in the human genome, not detected in entire dataset (grey). Identification of GOCC terms using DAVID[95]. **c** Venn diagram displaying proteins identified in current study alongside results of single-fibre papers published to date. Due to differences in experimental design, and to make valid comparisons, ref. 29 were filtered for young participants with <30% missing values

fibres (Fig. 1f). Labelled points indicate proteins that were uniquely differentially expressed within our study (compared with refs. 29,30; for a full listing, please see Supplementary Data 1-Tab 8). Our results are similar to ref. 30, which identified 232 differentially expressed proteins between fibre types (reanalysed by us with the same imputation and empirical Bayes methods as our study). Unsupervised hierarchical clustering revealed a clear separation of fibre types, although, as expected, there was some biological variability amongst individuals (Fig. 1g).

Our analysis revealed contractile proteins that have not previously been identified as having fibre-type-specific expression profiles (Supplementary Data 1-Tab 8). These proteins included MYH3 and MYL4, which code for myosin heavy and light chains present in embryonic skeletal muscle; their presence in adult muscle is usually interpreted as a sign of muscle regeneration, but little is known about these two members of the myosin family in the context of exercise[39]. We also observed many of the expected differences between fibre types. Type I fibres were characterised by a greater abundance of contractile proteins belonging to the myosin heavy (MYH7) and light (MYL6B) chains, as well as troponin (TNNT1, TNNC1, TNNI1) and an isoform of the sarcoplasmic reticulum $Ca^{2+}$-ATPase (ATP2A2, also named SERCA2[40,41];), while type II fibres possessed a greater abundance of MYH2, MYL1, TNNT3, TNNC2, TNNI2 and also ACTN3.

Most studies on muscle fibre heterogeneity have focused on contractile proteins[33,42]. However, as confirmed in our study, the diversity between muscle fibres is not only restricted to contractile proteins but extends to a wide range of subcellular systems - including ionic transport, cellular calcium signalling and metabolism[42]. Approximately 30% of the proteins identified as differentially expressed between fibre types were mitochondrial—with most of these mitochondrial proteins (~90%) having a greater abundance in type I fibres. These differentially expressed mitochondrial proteins were enriched in pathways associated with fatty acid oxidation, as well as other proteins involved in the electron transport chain (Supplementary Data 1-Tab 7).

## Influence of mitochondrial content on fibre-type-specific expression profiles

Of the 391 mitochondrial proteins quantified in the present study, 314 (~70%) were also quantified in the previous two single-fibre proteomic studies (Fig. 2a, Supplementary Data 2-Tab 1). The top 5 GO Biological Process (BP) terms for these commonly detected mitochondrial proteins were related to mitochondrial electron transport (NADH to ubiquinone and respiratory chain complex I assembly), the tricarboxylic acid cycle, fatty acid beta-oxidation and mitochondrial ATP synthesis coupled proton transport (Fig. 2b, Supplementary Data 2-Tab 2). Many of these mitochondrial proteins had a significantly greater abundance in type I fibres (Fig. 2c—top panel, Supplementary Data 1-Tab 7), which is consistent with the superior capacity of type I fibres for mitochondrial oxidative phosphorylation and fatty acid oxidation[43].

and ref. 30 were filtered for proteins with <30% missing values. **d** Proteins identified by three single-fibre studies in (**c**) underwent GOCC analysis. Top 5 terms of 844 proteins in intersect of proteins identified by Deshmukh et al., present study and Murgia et al. are presented. Top terms and corresponding $\log_{10} P$ values identified using DAVID[95] and computed by one-sided Fisher's Exact Test. **e** MDS plot shows distances among fibre-type pools for PRE exercise type I (blue, $n = 16$) and type II (red, $n = 16$) samples. **f** Volcano plot illustrating fold-change and significance of proteins in PRE samples between type I ($n = 16$) and type II ($n = 16$) fibres. Points were coloured if adjusted $P < 0.05$ by limma analysis and a Log2 Fold Difference of >0.2 (higher in Type II) or <−0.2 (higher in Type I). Points labelled if differentially expressed uniquely within our study (compared with refs. 29,30). **g** Heatmap displaying z-score values of differentially expressed proteins comparing type II ($n = 16$) to type I ($n = 16$) fibre pools within PRE samples. Samples cluster by fibre type I (blue) or type II (red).

An important question that has not been adequately addressed is whether these fibre-type-specific expression profiles for mitochondrial proteins are independent of the greater mitochondrial content in type I versus type II human skeletal muscle fibres[13,44,45]. To examine the potential influence of mitochondrial content on fibre-type-specific expression profiles observed for mitochondrial proteins, we first calculated the mitochondrial protein enrichment (MPE)—a value that identifies the contribution of mitochondrial protein intensities to the overall detectable proteome[8]. Consistent with previous research[30], there was a greater abundance of proteins associated with the mitochondria cellular component in type I compared to type II fibres (Fig. 2d, Supplementary Data 2-Tabs 3–5). As this supported an overall greater content of mitochondria in type I versus type II fibres in our participants, we next employed our previously described mitochondrial normalisation strategy[8], which aims to correct for the bias introduced by differences in total mitochondrial content. This allowed us to compare the abundance of individual mitochondrial proteins relative to the mitochondrial proteome in each fibre type (i.e. mitochondria-corrected relative abundance).

Post normalisation, less than one-third of the mitochondrial proteins we had observed to have a greater abundance in type I fibre pools remained differentially expressed (compare Fig. 2c—bottom panel, Supplementary Data 2-Tabs 6, 7, with Fig. 2c—top panel, Supplementary Data 1-Tab 7). This analysis suggests that many of the differences in mitochondrial protein abundances between fibre types reported by us, and others[29,30], might be due to differences in total mitochondrial content rather than fibre-type-specific remodelling of the mitochondrial proteome. With this approach, we also identified nine additional mitochondrial proteins with a greater abundance in type II than type I fibres; these proteins included ALDH1B1, SDHA, RIDA, SLC25A12, GRSF1, PARK7, HSPE1, GPX4, PNPO (Fig. 2c). None of these nine mitochondrial proteins had previously been reported to be differentially expressed in type II versus type I muscle fibres, and further research is required to explore their potential role in functional differences between the fibre types.

## Proteomic responses to different types of training in type I and type II skeletal muscle fibres

One published proteomic study has examined fibre-type-specific adaptations to one type of exercise training (1 h at 75–90% of maximum heart rate, 4x/week for 12 weeks) in human skeletal muscle[30]. However, no study has investigated if there are fibre-type-specific changes to the proteome that depend on the nature of the exercise stimulus. The size principle states that type I fibres are mostly utilised at lower exercise intensities, whereas type II fibres are increasingly recruited at higher exercise intensities[17,21]. Therefore, a long-standing hypothesis is that training-induced changes to the proteome in response to low-intensity exercise will mostly be confined to the type I fibres that are predominantly recruited with this type of exercise[19,46]. In contrast, higher-intensity exercise (e.g. all-out sprinting) has been

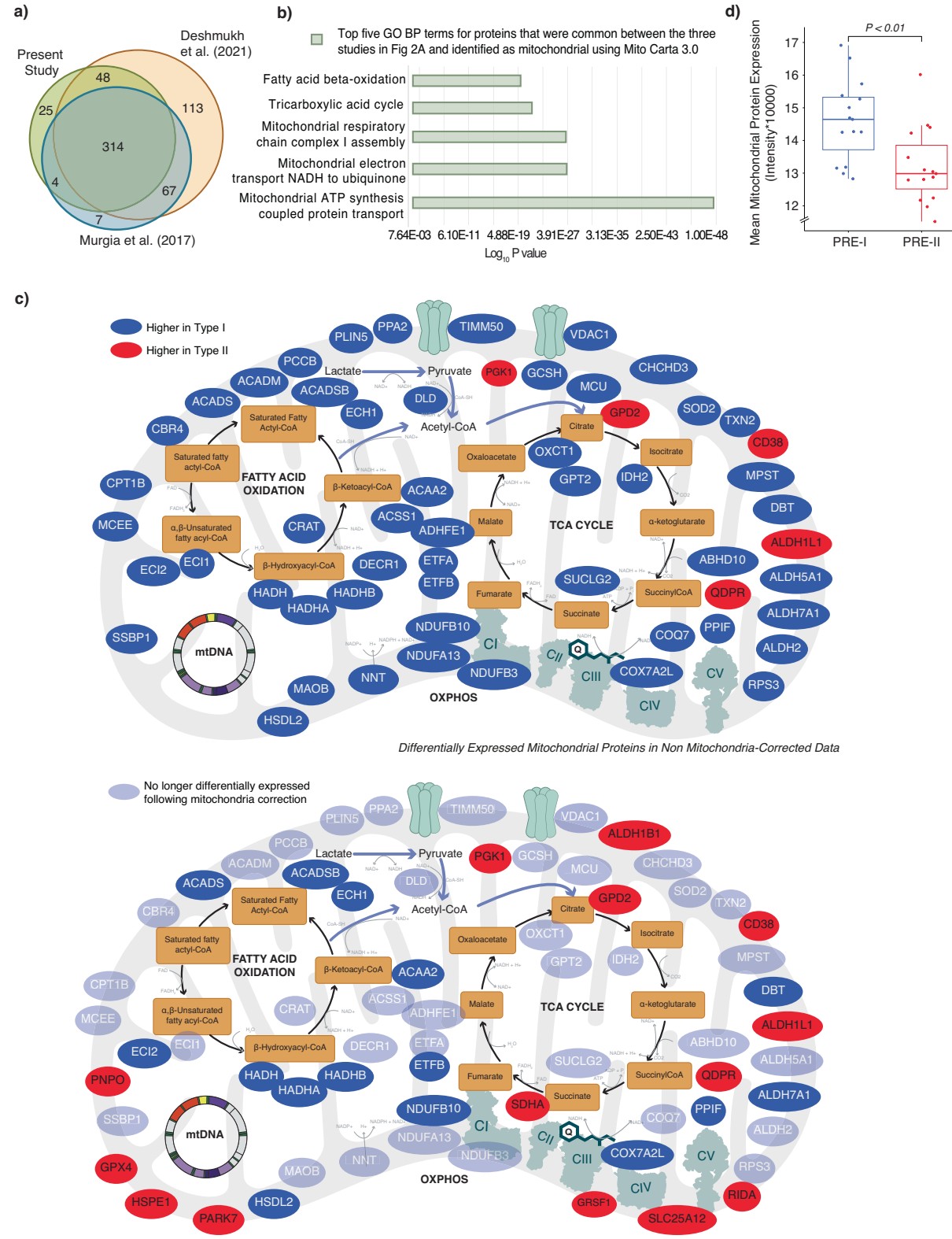

*Differentially Expressed Mitochondrial Proteins in Non Mitochondria-Corrected Data*

*Differentially Expressed Mitochondrial Proteins in Mitochondria-Corrected Data*

proposed to stimulate greater adaptations in type II fibres compared to exercise performed at lower intensities[47].

To investigate potential fibre-type-specific changes to the proteome in response to divergent exercise stimuli, we analysed the data from 16 healthy young men who trained 3 to 4 times per week for 8 weeks after random assignment to one of two very different types of training – MICT or SIT (Fig. 3a). By design, both exercise groups had very different exercise prescription characteristics; there was a 4.7-fold greater exercise intensity for SIT and a 5.2-fold greater training volume for MICT (Fig. 3b). The divergent exercise prescriptions, and their associated effects on the metabolic response to exercise, can be observed by the larger increase in blood lactate concentration following SIT (13-fold; $1.4 \pm 0.3$–$18.6 \pm 3.5$ mmol·L$^{-1}$; $P < 0.001$) when compared to MICT (1.6-fold;

**Fig. 2 | Analysis of the mitochondrial proteome reveals fibre-type differences in the pre-training samples. a** A Venn diagram displaying the number of mitochondrial proteins annotated according to the mitochondrial protein database Mitocarta 3.0[95] in the present study, alongside two published human single-fibre datasets (refs. 29,30). Only proteins with less than 30% missing data pre-intervention were annotated to Mitocarta 3.0. **b** Top gene ontology biological processes (GOBPs) in which the identified mitochondrial proteins common to all three studies illustrated in (**a**) are mainly involved. The bars represent $Log_{10}P$ values, where $P$ represents the significance of each GOBP enriched by the mitochondrial proteins. The $P$ value was computed using the DAVID software[95] and computed by a one-sided Fisher's Exact Test. **c** Illustration of mitochondrial proteins that were differentially expressed between type I and type II fibres in the PRE samples following standard normalisation (Upper panel) and proteins that were identified as differentially expressed when the abundance of individual mitochondrial proteins was expressed relative to the total mitochondrial protein abundance of each fibre type (i.e. mitochondria-corrected relative abundance[8]) (Lower Panel). Proteins that are no longer differentially expressed following mitochondrial normalisation are shaded light blue. Proteins were labelled as mitochondrial according to Mitocarta 3.0[95]. **d** Comparison of Mean Mitochondrial Protein Expression in the PRE exercise samples for type I (blue) ($n = 16$) and type II fibres (red) ($n = 16$). Replicates are biological. The blue and red boxes indicate the median and the 25th/75th percentile with the whisker lines representing the distribution extending to 1.5 of the interquartile range. An adjusted $P$ value of 0.0015 was computed with a two-tailed paired t-test.

---

$1.38 \pm 0.43$–$2.18 \pm 0.38$ mmol·L$^{-1}$; $P = 0.001$) (Fig. 3b; Supplementary Data 3-Tab 1). Similarly, blood pH levels were only decreased following SIT (from $7.35 \pm 0.03$–$7.02 \pm 0.09$; $P < 0.001$) and not MICT (from $7.35 \pm 0.02$–$7.37 \pm 0.02$; $P > 0.05$) (reflected by fold differences in [H$^+$] in Fig. 3b). These results agree with the well-described effect of exercise intensity on increasing glycolytic flux through anaerobic pathways and the associated increase in lactate and decrease in pH[3,48]. Immunohistochemical analysis of muscle sections confirmed a greater depletion of muscle glycogen in type I than type II fibres following the first MICT session, and greater muscle glycogen depletion in the type II fibres following SIT versus MICT (Fig. 3c). This is consistent with previous research[49,50] and demonstrates that the two types of training employed were associated with very different skeletal muscle fibre recruitment patterns (i.e. MICT recruited predominantly type I fibres while SIT recruited both type I & type II fibres; see the schematic in Fig. 3d).

When correcting for multiple comparisons (Fig. 4a)[51], we did not detect any significant changes in the abundance of individual proteins in either fibre type following either training type (Fig. 4b; Supplementary Data 4). Our results are consistent with those of the only published single-fibre proteomic study utilising exercise[30], which only reported changes in the abundance of individual proteins when they analysed their data with a posteriori information fusion scheme that combines fold change and statistical significance (unadjusted $P$ values from thousands of individual paired t-tests) (Fig. 4c)[52]. There is, however, good evidence that empirical Bayes methods better control for false positives when analysing large datasets[53]. Indeed, when we re-analysed the data of ref. 30 with the same imputation and empirical Bayes methods as our study (i.e. limma[51]), we observed no significant changes in protein abundance in either fibre type following their training programme (Fig. 4d; Supplementary Data 4). Another notable difference between the two studies is that the fold changes reported by Deshmukh et al.[30] ($-4.5$ to $3.6$ Log$_2$FC; Fig. 4c) were much larger than those observed in our study ($-1.5$ to $1.7$ Log$_2$FC; Supplementary Data 4). These larger fold changes may partially be due to differences in quantification methods between our two studies but would have contributed to the greater number of differences identified via the posteriori information fusion scheme. Thus, our results raise important questions about the ability of human studies with small to moderate sample sizes to confidently detect training-induced changes of individual proteins in single-fibre pools, especially when making appropriate adjustments for multiple hypothesis testing. This can be attributed, at least in part, to the inherent inter-individual/inter-fibre variability of human muscle biopsy samples and the individual variability in the response to training.

Another interesting observation was that there were fewer differentially expressed proteins between type I and type II fibres post-training compared to pre-training (compare Fig. 5a, b, Tables S5-Tabs 1, 2, with Fig. 1f). When comparing training types, more differentially expressed proteins were higher in both type I and type II fibre pools following MICT than SIT (Fig. 5c; Supplementary Data 5-Tab 3). Additionally, more than 70% of the proteins identified as differentially

expressed between type I and type II fibres pre-training were not significantly different post-training (Fig. 5d; Supplementary Data 5-Tab 4). This is consistent with previous reports that type II fibres become more like type I fibres with endurance training[54–56]. We add that this reduced difference between type I and type II fibres post-training was more pronounced for SIT (greater recruitment of both type I and II fibres; see Fig. 3c, d) than MICT. There were, however, an additional 23 differentially expressed proteins between the two fibre types post-MICT that were not differentially expressed between the two fibre types pre-training (Fig. 5d; Supplementary Data 5-Tab 4). This included Cytochrome b5 reductase (CYB5R3), which was higher in type I fibres following MICT and has recently been linked to the regulation of lipid metabolism and modest lifespan extension in mice[57].

Given the challenges in confidently detecting training-induced changes of individual proteins in human single-fibre pools, we turned our attention to protein groups within different pathways; this allowed for an overview of protein changes within pathways (Fig. 5e; Supplementary Data 5-Tab 5). Network analysis conducted on differentially expressed proteins in the two fibre types after both training interventions (i.e. MICT and SIT) demonstrates that differences were more common between the same fibre type than the same exercise training type (Fig. 5f; Supplementary Data 5-Tab 6). For example, for both types of training, striated muscle contraction proteins associated with slow-twitch skeletal muscle isoforms were predominately higher post-training in type I fibres (TNNT1, TNNI1, TNNC1, TPM3, MYL2 and MYL3), while proteins associated with fast-twitch skeletal muscle isoforms were predominately more abundant post-training in type II fibres (TNNC2, MYH6, TNNT3, MYBPC2, TPM1, TNNI2, MYL1 and MYL4). The network analysis also indicated that proteins associated with glycolysis were mostly differentially expressed only in type II fibres after both types of training; proteins belonging to the glycogen synthesis pathway were only observed to be greater in type II fibres post-MICT and not post-SIT. Finally, although fatty acid oxidation pathway proteins were predominately differentially expressed in type I fibres following training, these differences were all present in the PRE samples.

## Fibre-type-specific effects of different types of exercise training on mitochondrial proteins

Mitochondrial content is greater in type I compared with type II fibres[58,59] (see also Fig. 2d). However, while research has reported that markers of mitochondrial content increase with exercise training[6,60–63], little is known about the fibre-specific effects of different types of exercise training. Consistent with our morphological (mitochondrial volume density obtained from TEM; Fig. 6a, b) and biochemical (citrate synthase [CS] activity; Fig. 6c) markers of mitochondrial content (Supplementary Data 6-Tab 1), we observed a significant increase in the mean abundance of known mitochondrial proteins only following MICT and not SIT (Fig. 6d) (Supplementary Data 6-Tab 2). This can likely be attributed to the five-fold greater volume of training completed by the MICT group[6,64], although a consensus has yet to be reached on the relative importance of exercise intensity versus volume

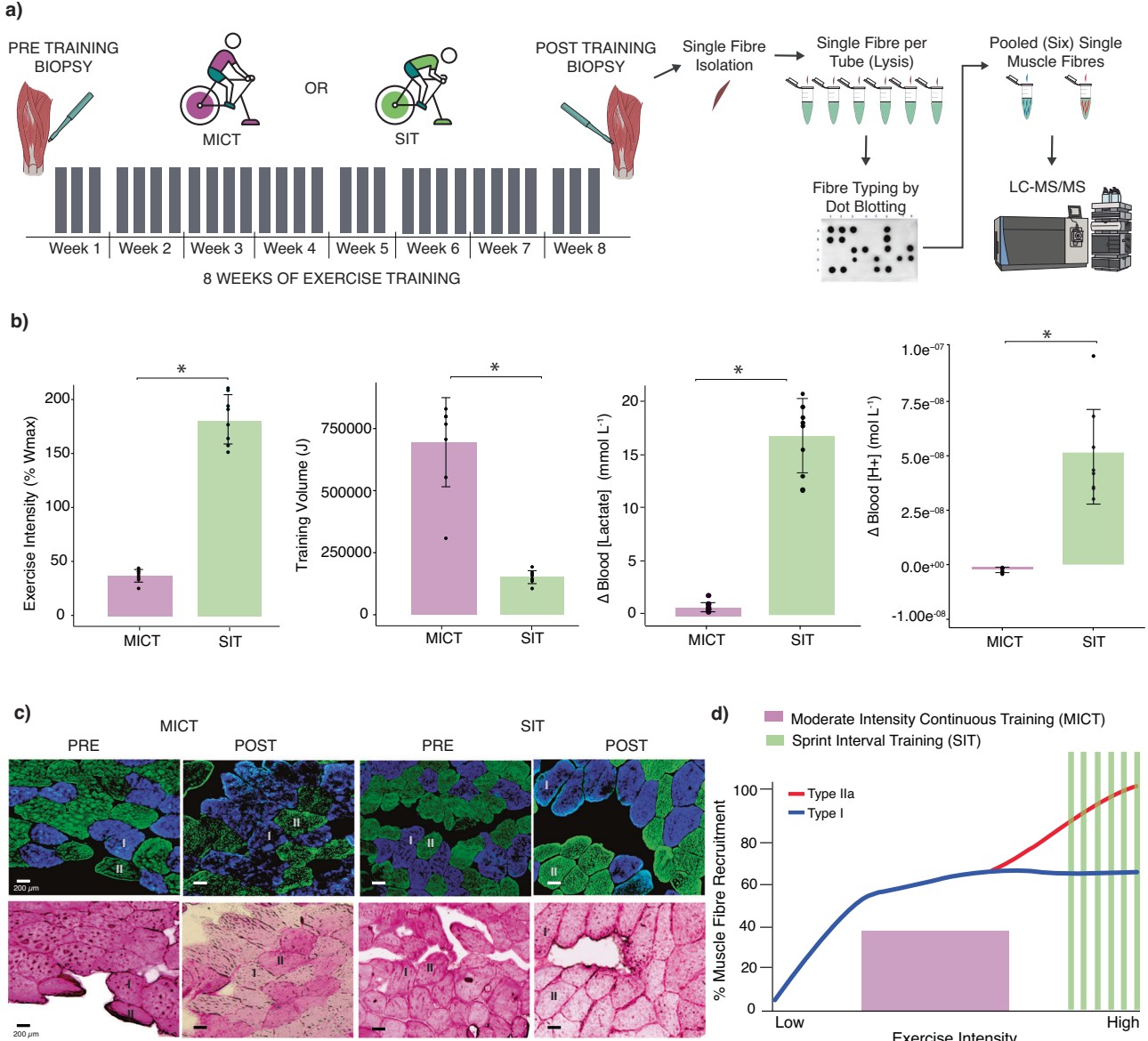

**Fig. 3 | Responses to moderate-intensity continuous training (MICT) and sprint interval training (SIT). a** 8-week exercise training regime for MICT and SIT as per described methodology. Skeletal muscle biopsies were taken before week 1 and after week 8. Grey bars indicate number of training sessions per week. Biopsies were taken at rest before first session and 72 h post final session. Single fibre lysates were fibre typed via dot-blotting and subsequently pooled by fibre type. **b** Bar graphs displaying fold differences between MICT ($n = 8$) and SIT ($n = 8$) for exercise intensity (%$\dot{W}_{max}$) and total training volume (J), with adjusted $P$ values of $8.1 \times 10^{-11}$ and $6.9 \times 10^{-7}$, respectively, computed by two-tailed t-tests. Bar graphs displaying the change in blood lactate concentration (mmol•L$^{-1}$) and hydrogen ion (H$^+$) concentration (mol•L$^{-1}$) from pre to post exercise following the first MICT ($n = 8$, $n = 7$ [H$^+$]) and the first SIT ($n = 8$) session, with adjusted $P$ values of $1.7 \times 10^{-9}$ and $1.6 \times 10^{-5}$, respectively, computed by two-tailed t-tests. Data are presented as mean ± SD; * adjusted $P$ value < 0.05. Individual points signify individual participant values. Antecubital venous blood was drawn at rest and immediately post exercise. **c** Representative images of the effects of a single session of exercise on fibre-type-specific glycogen depletion patterns in human vastus lateralis muscle samples collected before and after MICT ($n = 4$) or SIT ($n = 4$). Top panels display immunofluorescent fibre-type staining for myosin heavy chain content, with type I muscle fibres (MHC7) identified as blue and type II fibres (MHC2) identified as green. Bottom panels display serial sections stained for glycogen (periodic acid-Schiff staining); a decrease in staining saturation reflects lower glycogen concentration. Bars = 200 μm. Each staining was repeated independently for 8 different samples. **d** Illustration of proposed hierarchal recruitment pattern of type I and IIa skeletal muscle fibres in relation to exercise intensity. MICT is in purple and SIT in green, with type I fibres and type II fibres in blue and red, respectively.

on training-induced changes in markers of mitochondrial content[62,65]. Changes in the mean abundance of known mitochondrial proteins following MICT occurred in both the type I and type II fibres (Fig. 6d); this suggests that there is recruitment of type II fibres, along with type I fibres, during long-duration, moderate-intensity exercise. There was a significant decrease in known mitochondrial protein abundance in type I fibre pools following SIT (Fig. 6d).

Given the training-specific changes in the mean abundance of known mitochondrial proteins, we next investigated training-induced changes in the total abundance of proteins in different mitochondrial functional groups in both fibre types. We observed fibre-type-specific adaptations of mitochondrial proteins to training (Supplementary Fig. 2a, Supplementary Data 8-Tab 1), consistent with the only other proteomic study investigating the effects of exercise training in pools of single fibres[30]. Our results suggest that the overall mean abundance of many mitochondrial functional groups is enhanced by high-volume, moderate-intensity, endurance training. Interestingly, the total abundance of TCA cycle, beta-

a)
Methodology used to compare Reisman et al. 2024
and Deshmukh et al 2021 for MICT Type I PRE vs POST Samples.

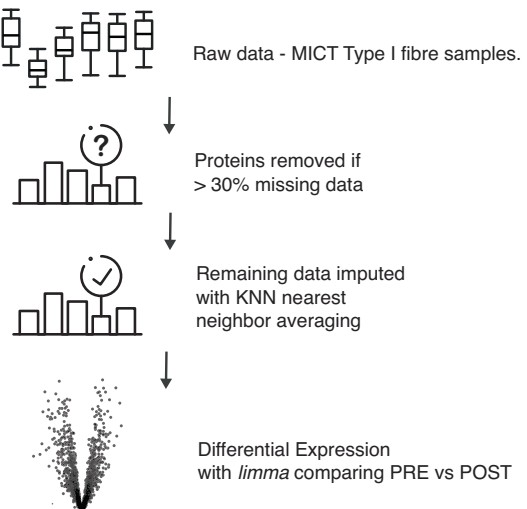

b) Significantly identified proteins after imputation alterations and
adjusted p values set to adj. p < 0.05 in Reisman et al. 2024

● Not Significant P > 0.05
● Only Significant P < 0.05, Not significant when adjusted.

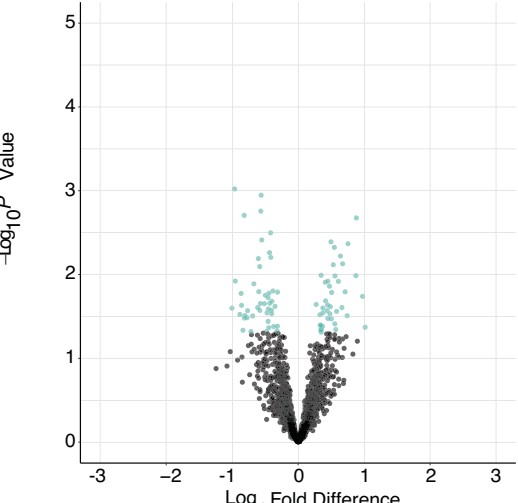

c) Original data from Deshmukh et al. 2021
showing proteins identified as significant with fusion factor.

● Not Significant P > 0.05
● Only Significant with fusion factor < 0.05

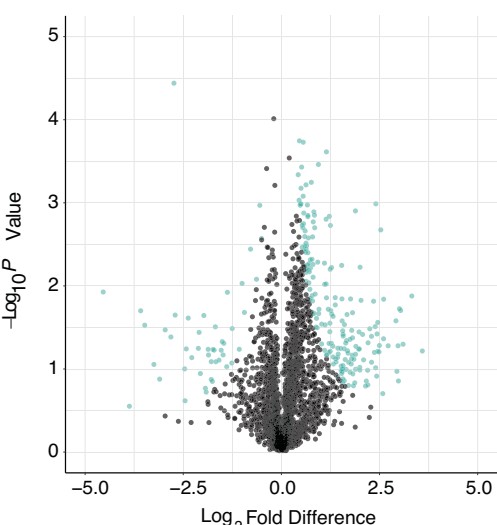

d) Significantly identified proteins after imputation alterations and
adjusted p values set to adj. p < 0.05 in Deshmukh et al. 2021

● Not Significant P > 0.05
● Only Significant P < 0.05, Not significant when adjusted.

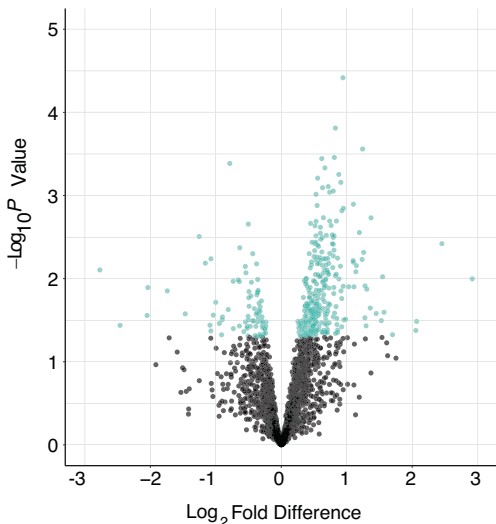

**Fig. 4 | Assessing the fold-change distribution of proteins within type I fibres following training. a** Methodology used to compare the results of the present study with those of ref. 30 in type I samples PRE vs POST moderate-intensity continuous training (MICT). The methodology shows that proteins were removed from the raw data if there was more than 30% missing data across the total matrix; this was followed by imputation with KNN nearest neighbour averaging with differential expression comparisons. **b** Volcano plot illustrating the fold-change distribution of proteins within type I fibres following MICT for the present study. Proteins are coloured in teal based on whether the proteins were differentially expressed with an unadjusted *P* value of <0.05. When the values were adjusted using the Benjamini-Hochberg method for multiple hypothesis testing, none of these proteins remained significant. **c, d** Volcano plots illustrating the fold-change distribution of proteins within type I fibres following MICT for Deshmukh et al. (**c, d**). Proteins are coloured in teal based on whether the proteins were differentially expressed with an unadjusted *P* value of <0.05 (**c**) or significant with a fusion factor of <0.05 (ref. 52) (**d**); when the values in (**c**) were adjusted using the Benjamini-Hochberg method for multiple hypothesis testing, none of these proteins remained significant.

oxidation and mitochondrial ribosome proteins increased in both fibre types from PRE to POST MICT (Fig. S2b, c, Supplementary Data 8-Tab 2). This latter result is similar to those of another proteomic study, which revealed changes in mitochondrial ribosomal proteins in whole-muscle samples following interval training[66]. These authors hypothesised that translational level regulation is a predominant factor controlling mitochondrial biogenesis in humans in response to exercise training. No changes in mitochondrial ribosome proteins were observed in either fibre type for SIT. This suggests the volume of MICT was sufficient to induce significant mitochondrial protein synthesis in both fibre types, whereas SIT may not provide a sufficient stimulus to significantly alter mitochondrial protein synthesis[67], and is consistent with our TEM findings (Fig. 6b).

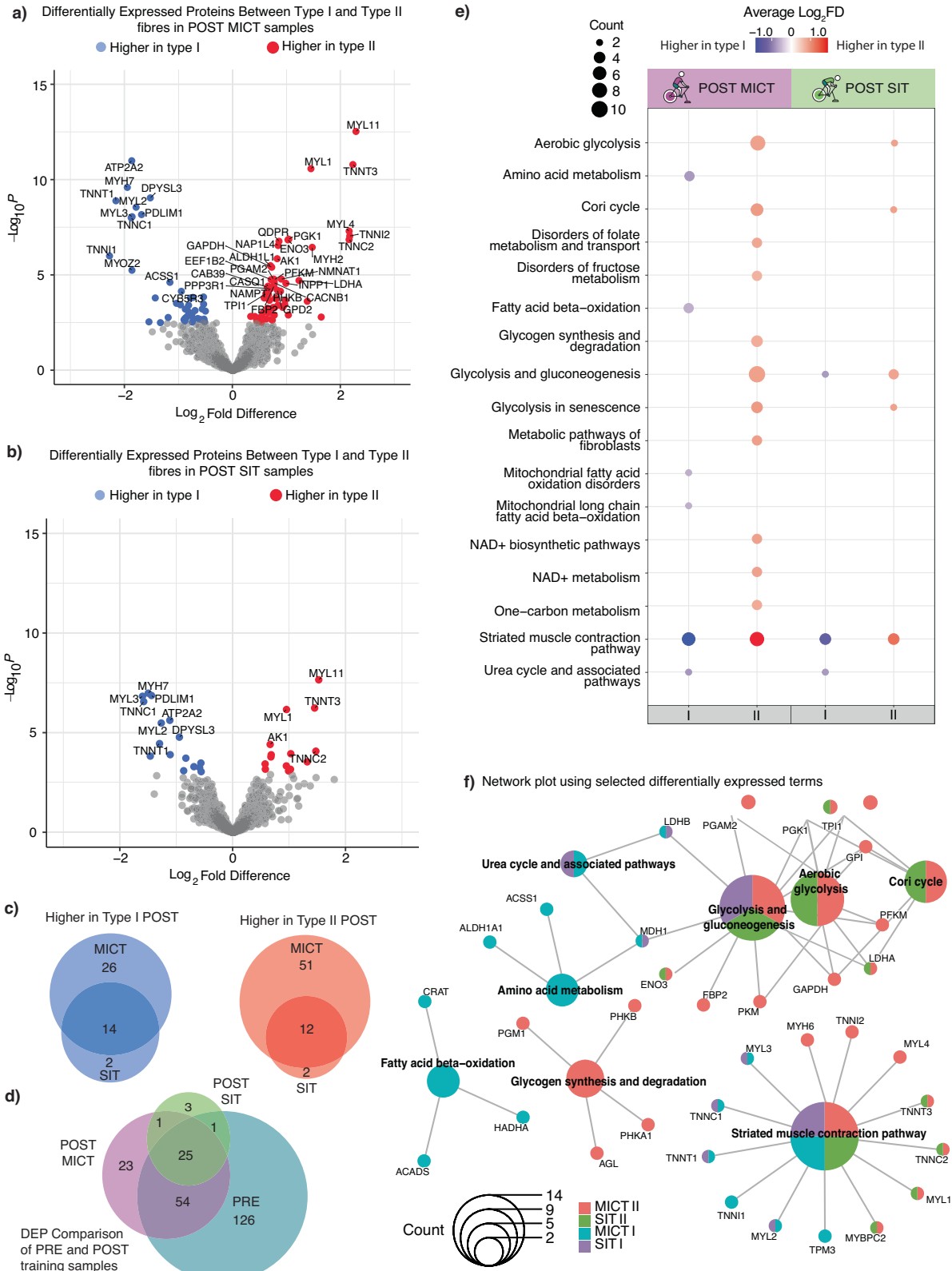

**a)** Differentially Expressed Proteins Between Type I and Type II fibres in POST MICT samples

**b)** Differentially Expressed Proteins Between Type I and Type II fibres in POST SIT samples

**c)** Higher in Type I POST / Higher in Type II POST

**d)** DEP Comparison of PRE and POST training samples

**e)** Average Log₂FD

**f)** Network plot using selected differentially expressed terms

## Fibre-type-specific expression profiles for mitochondrial proteins independent of training-induced changes in mitochondrial content

Given the greater changes in mitochondrial content that occur following MICT compared with SIT[61,68–69]; (see also Fig. 6d), we turned our attention to investigating if fibre-type-specific changes in different mitochondrial functional groups with training were independent of changes in our marker of total mitochondrial content (i.e. MPE). Once again, we employed our normalisation strategy[8] to remove the bias introduced by the changes in mitochondrial content in each fibre type following both types of training (Fig. 7a). This enabled us to investigate differences in mitochondrial protein abundances between the two fibre types without the bias introduced by training-induced changes in total mitochondrial content (Supplementary Data 7-Tabs 1, 2). Post

**Fig. 5 | Differentially expressed proteins between type I and type II fibres post-training. a** Volcano plot illustrating distribution of proteins differentially expressed between type I ($n = 8$) and II ($n = 8$) fibres in POST MICT samples. Points were labelled based on whether they were differentially expressed, according to adjusted $P$ values from the two-sided limma analysis. **b** Volcano plot illustrating distribution of proteins that were differentially expressed between type I ($n = 8$) and II ($n = 8$) fibres in the POST SIT samples. Points were labelled based on whether they were differentially expressed, according to adjusted $P$ values from the two-sided limma analysis. **c** Venn Diagrams displaying number of proteins identified as higher in type I or type II fibres for both exercise types, with proteins that appear in both datasets seen in the overlap. **d** Venn Diagram displaying number of proteins that were differentially expressed between type I and type II fibres in the PRE samples ($n = 16$ participants), compared with number of differentially expressed proteins between type I and type II fibres in the POST MICT ($n = 8$) and SIT ($n = 8$) samples.

**e** EnrichPlot dot plot of protein expression in type I vs type II fibres for the POST MICT ($n = 16$) and SIT ($n = 16$) samples. Pathways were identified using WikiPathways. Top 15 terms, according to adjusted $P$ values (<0.05), are displayed. Red dots identify pathways higher in type II fibres, blue dots indicate pathways higher in type I fibres. The saturation of each colour is proportional to the average Log2 fold difference for each exercise and fibre group. Size of dot indicates how many proteins correspond to each pathway. For all proteins, the corresponding $\log_2$ fold change and $P$ values from the differential expression analysis were used. All terms can be found in Supplementary Data 5-Tab 5. **f** ClusterProfiler network plot using differentially expressed (Benjamini-Hochburg adjusted $P < 0.05$) terms. Pathways were identified using WikiPathways. Colours indicate in which exercise and fibre group the proteins in that WikiPathway term were found. Size of dot indicates how many proteins corresponded to each pathway. Displayed terms were selected for readability. All terms can be found in Supplementary Data 5-Tab 6.

normalisation, very few training-induced changes in mitochondrial protein abundances remained in either fibre type, following either training intervention. One exception was the relative decrease in the abundance of complex IV subunits in both type I and II fibres following SIT (Fig. 7b, c). Thus, when changes in mitochondrial content are controlled for, very-high-intensity SIT appears to deprioritise the biogenesis of OXPHOS components involved in the final step of the mitochondrial electron transport chain (i.e. complex IV). The other exception was the increase of proteins associated with fatty acid oxidation in type I fibres following MICT, which our results suggest was greater than the overall training-induced increase in mitochondrial content (Fig. 7d). This adds to the evidence that MICT provides a powerful stimulus to increase fatty acid oxidation[70,71]; we add that this adaptation occurs predominantly in type I skeletal muscle fibres (with the exercise stimulus employed in our study).

## Discussion

Proteomics applied to human skeletal muscle remains in its infancy but has introduced new opportunities to explore the complexity of the biological networks underlying fibre-specific differences and responses to exercise training. In the present study, we developed a sensitive MS-based proteomics workflow that allowed for the pooling of single muscle fibres, and the utilisation of TMT labelling, to accommodate our complex experimental design and increase the quantitative accuracy of our results. In addition, we incorporated a larger sample size than previous studies to better account for the individual variability in human skeletal muscle and to increase the robustness of our findings. Consistent with previous research[30], we identified more than 200 differentially expressed proteins between type I and type II fibres at baseline (Fig. 1f). However, further analysis of these results allowed us to make two important observations. The first is that less than half of mitochondrial proteins (20 of 56) with fibre-type-specific expression profiles remained significantly different when we used our published normalisation strategy to account for the overall greater content of mitochondria in type I versus type II fibres (Fig. 2c−bottom panel). Thus, many of the differences in mitochondrial protein abundances between fibre types reported by us (Fig. 2c−top panel), and others[29,30], are likely due to differences in total mitochondrial content rather than representing a fibre-type-specific mitochondrial proteome. A further finding is that more than 70% of the proteins identified as differentially expressed between type I and type II fibres pre-training were not significant post-training (Fig. 5d; Supplementary Data 5-Tab 4). This highlights the challenge of distinguishing between protein expression profiles intrinsic to different fibre types and those that are a consequence of adaptations to habitual physical activity (or inactivity).

Repeated muscle contractions, via habitual physical activity or exercise training, provide a potent stimulus to alter the expression of proteins and confer many health benefits[4]. The development of mass-spectrometry-based proteomics has advanced our understanding of training-induced changes in protein abundance in mixed-fibre

samples[8,72–74]. However, while it is well known that adaptations to training depend on the characteristics of the exercise stimulus[10], no proteomic study has investigated the effects of different exercise stimuli on fibre-type-specific adaptations. Therefore, an important feature of our study design was to investigate fibre-type-specific adaptations in response to two types of training that were very different in volume, intensity, metabolic stress and fibre recruitment patterns (Fig. 3). Although our study design was motivated by the many changes reported as significant in the only published single-fibre proteomic study[30], we did not detect any significant changes in the abundance of individual proteins in either fibre type following either training type. Subsequent re-analysis of the previously published data highlighted two potential contributing factors−the greater fold differences reported in the previous study (which may be related to differences in protein quantification method, and/or the choice of imputation method[75]) and the determination of statistical significance via a posteriori information fusion scheme that combines fold change and statistical significance (Fig. 4c; Supplementary Data 4). These results collectively highlight some of the methodological challenges that need to be overcome to confidently detect training-induced changes of individual proteins in single-fibre pools - especially given the inherent inter-individual and inter-fibre variability of human muscle biopsy samples.

Given the well-known effects of different types of training on mitochondrial content (see also Fig. 6), a further important feature of our study was to investigate whether the fibre-type-specific changes observed by us for the total abundance of proteins in different mitochondrial functional groups were independent of training-induced changes in total mitochondrial content. To do this, we used our previously described normalisation strategy to control for the bias introduced by changes in mitochondrial content in each fibre type following both types of training[8]. This analysis revealed that most training-induced changes in mitochondrial protein abundances were no longer significant when normalised to the overall increase in mitochondrial content (compare Fig. 7 with Supplementary Fig. 2). However, a surprising observation was the decrease in the abundance of complex IV subunits, relative to the overall increase in mitochondrial protein content, in both type I and II fibres following SIT. It has been hypothesised that complex IV downregulation may be a consequence of allosteric feedback inhibition by ATP[76]. This deprioritisation of the OXPHOS components involved in the final step of the mitochondrial electron transport chain is intriguing given that when also controlling for changes in mitochondrial content, we have previously reported greater changes in mitochondrial respiration following SIT than MICT[69]. The results of our normalisation strategy also revealed a discordant regulation of proteins associated with fatty acid oxidation, with an increase of these proteins in type I but not type II fibres following MICT. A previous study has also reported greater increases in perilipin (PLIN) 2 and 5 in type I compared with type II fibres following MICT[77]. These adaptations may facilitate greater

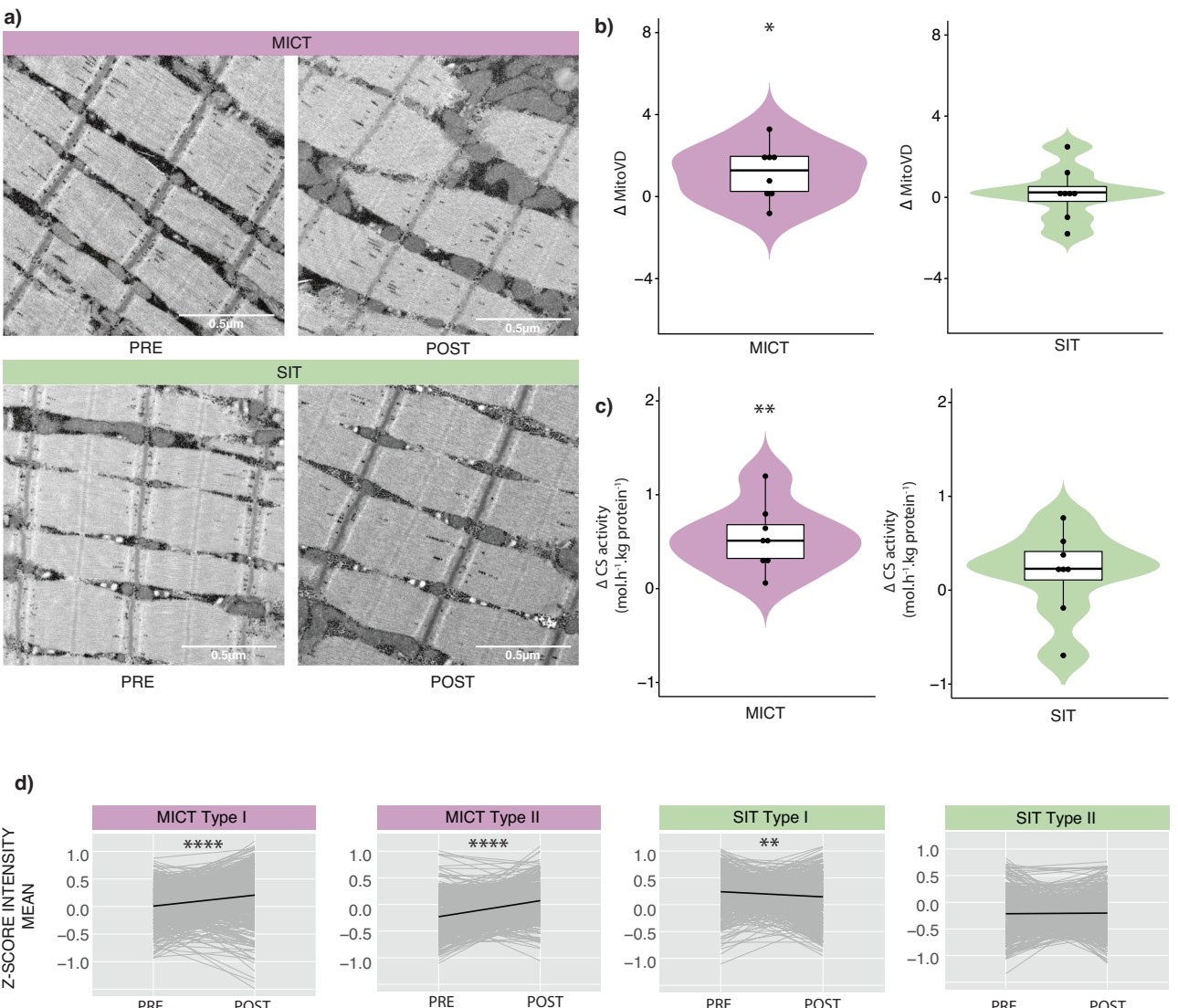

**Fig. 6 | Changes in markers of mitochondrial content with training.**
**a** Representative images of skeletal muscle samples analysed using transmission electron microscopy (TEM) pre- and post-training for participants who performed either moderate-intensity continuous training (MICT) or sprint interval training (SIT). Note the difference in the abundance of mitochondria post MICT. Bars = 0.5 μm. **b** Violin plots of the absolute change (Δ) in mitochondrial volume density (Mito$_{VD}$−proportion of the muscle volume occupied by mitochondria) from pre- to post-training following MICT (purple; $n = 8$) and SIT (green; $n = 8$). Twenty images were randomly obtained from at least five fibres of eight individuals of each training group and quantified PRE and POST training. The black boxes indicate the median and the 25th/75th percentile, with the whisker lines representing distribution extending to 1.5 of the interquartile range. Adjusted $P$ values of 0.018 (MICT) and 0.99 (SIT) were computed by two-tailed paired t-tests. **c** Violin plots of absolute change (Δ) in citrate synthase activity (CS; mol.h$^{-1}$.kg protein$^{-1}$) from pre- to post-training following MICT (purple, $n = 8$) and SIT (green, $n = 8$). Two samples from each individual (PRE and POST training) were measured in triplicate for Citrate Synthase activity. The black boxes indicate the median and the 25th/75th percentile, with the whisker lines representing distribution extending to 1.5 of the interquartile range. Adjusted $P$ values of 0.0034 (MICT) and 0.29 (SIT) were computed by two-tailed paired t-tests. **d** Scaled profile plots showing the relative abundance of mitochondrial proteins (grey) pre- and post-exercise in the MICT (purple, $n = 16$) and SIT (green, $n = 16$) groups for each fibre type, and the mean relative abundance (black) of the proteins in each group. Data are shown as Δ mean z-score. * indicates $P$ values for significance (*: $P < = 0.01$, **: $P < = 0.001$, ***: $P < = 0.0001$, ****: $P < = 0.00001$) by two-sided paired t-tests from pre- to post-training.

intramuscular triglyceride lipolysis and fatty-acid oxidation in type I fibres when metabolic demand increases[78,79].

In summary, our study incorporated several unique elements (a sensitive MS-based proteomics workflow, the largest sample size to date, two very different types of training stress and an in silico mitochondrial normalisation strategy) to investigate fibre-specific adaptations to different types of training. This provides an important resource to better understand how different fibre types adapt to different training interventions, with potential insights for how best to prescribe exercise to improve both health and human performance. In contrast to the decreased abundance of complex IV subunits in both type I and II fibres following SIT, and the increase in proteins associated with fatty acid oxidation only in type I fibres with MICT, most training-induced changes in mitochondrial protein abundances were no longer significant when normalised to the overall increase in mitochondrial content.

## Limitations of the study
This study recruited healthy, young, active men and, therefore, some of the conclusions may not apply equally well to individuals of different sexes, ages (e.g. the elderly), ethnicity, or health status. Another potential limitation is the use of pooled muscle fibre samples; while

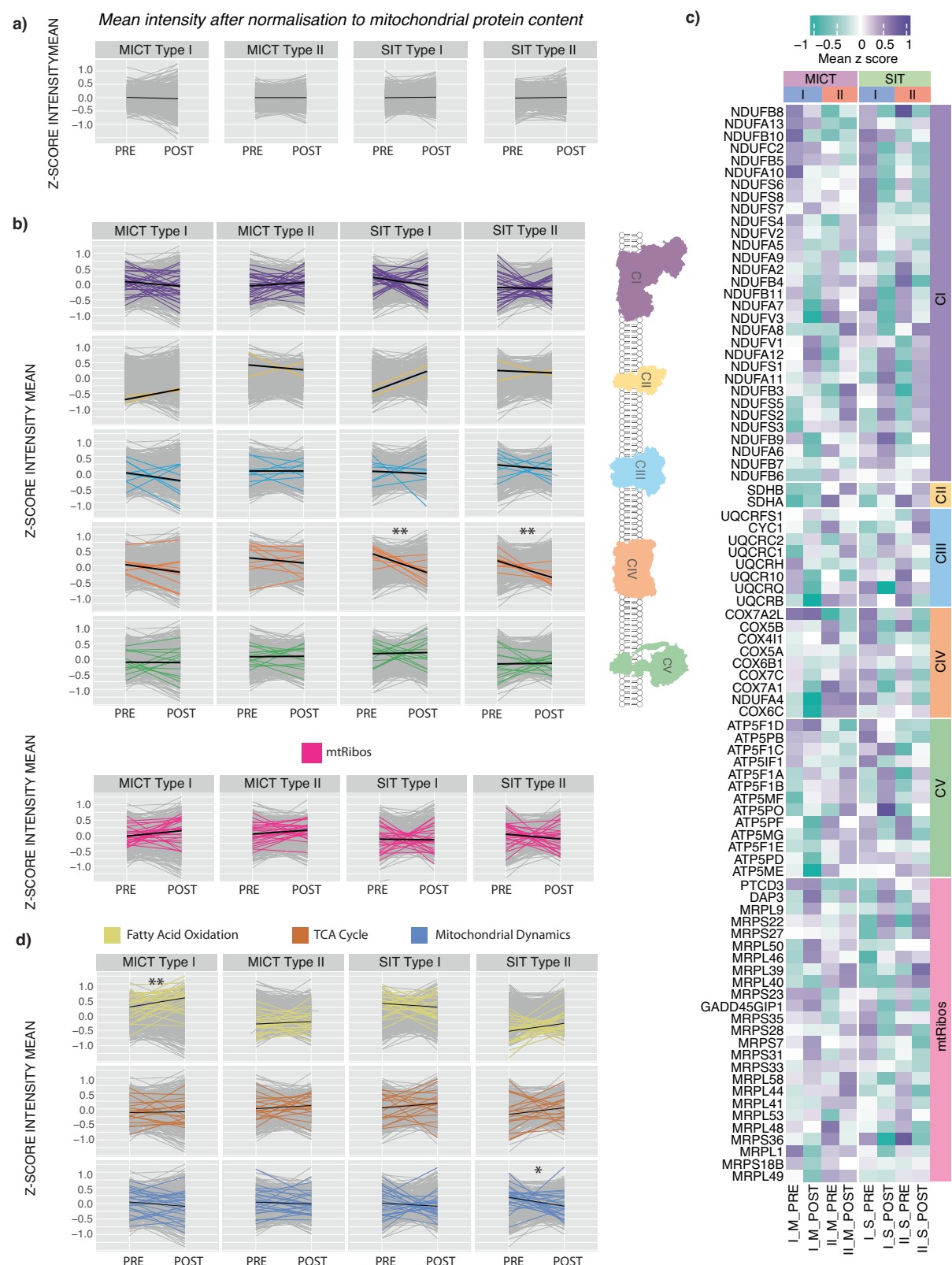

this assists in reducing biological variability amongst samples, it may not be a true representation of some single-fibre differences both pre- and post-training. Additionally, and similar to previous research[30], the type II fibre pools for some participants may have included a small proportion of type IIx and/or type IIa/x fibres[80,81]. Future research in this area should also consider analysing type IIx fibres, which may help explain some of the variability of the proteomic data in the type II fibre

pools reported by us and others. To better account for the inherent inter-individual/inter-fibre variability of human muscle biopsy samples, and the individual variability in the response to training, studies with even larger sample sizes should also be considered. It was not possible for us to determine if the fibre-specific adaptations to the two different types of exercise training had functional consequences, but it is unlikely that small changes in individual proteins are able to explain

**Fig. 7 | Normalisation to mitochondrial protein content reduces fibre-type differences in mitochondrial protein abundances. a** Scaled profile plots with the application of mitochondrial normalisation showing changes in the relative abundance of mitochondrial proteins (grey) pre- and post-training in the moderate-intensity continuous training (MICT) and sprint interval training (SIT) groups for each fibre type. The mean of each group is indicated by the black line. Data is shown as Δ mean z-score. **b** Scaled profile plots with the application of mitochondrial normalisation showing changes in the relative abundance of the subunits in each of the oxidative phosphorylation (OXPHOS) complexes (CI to CV), as well as the mitochondrial ribosomes, in type I and type II fibre types in response to both types of training (MICT, $n = 16$ and SIT, $n = 16$). The mean of each group (all proteins identified in each pathway using MitoCarta 3.0) is indicated by the black line. Data is shown as Δ mean z-score. * indicates $P$ value for significance (**: $P \leq 0.001$) based on

two-sided paired t-tests of the pre- to post-training values. **c** Heatmap displaying z-score values of protein subunits of the OXPHOS complexes, as well as the mitochondrial ribosome, in each of the two training groups (MICT and SIT), in type I and type II fibre type pools pre- and post-training, with the application of mitochondrial normalisation. **d** Scaled profile plots with the application of mitochondrial normalisation showing changes in the relative abundance of proteins involved in Fatty Acid Oxidation, the tricarboxylic acid cycle and mitochondrial dynamics (all proteins identified in pathway using MitoCarta 3.0) in type I and type II fibre type pools in response to both types of training (MICT, $n = 16$ and SIT, $n = 16$). The mean of each group is indicated by the black line. Data are shown as Δ mean z-score. * indicates $P$ value for significance (*: $P \leq 0.01$, **: $P \leq 0.001$) based on two-sided paired t-tests of the pre- to post-training values.

the observed improvements in exercise performance. Finally, it is important to note that there are several quantitative mass-spectrometry-based methods used to detect changes in the proteome, each with their advantages and disadvantages, and the results are not always directly comparable.

### Resource availability
**Lead contact.** Further information and requests for reagents and resources may be directed to and will be fulfilled by the lead contact, Professor David J Bishop (David.Bishop@vu.edu.au).

## Methods
### Experimental model and subject details
**Ethics approval.** Ethics approval for the study was obtained from the Victoria University Human Research Ethics Committee (HRE17-075) and conformed to the standards set by the latest revision of the Declaration of Helsinki. The study was registered as a clinical trial under Australian New Zealand Clinical Trials Registry (ANZCTR; ACTRN12617001105336).

**Statistics and reproducibility.** Twenty-eight healthy men initially volunteered to take part in this study. The data from sixteen participants ($27.5 \pm 5.2$ years; $177 \pm 7$ cm; $73.8 \pm 8.7$ kg; $23.6 \pm 2.6$ BMI; Supplementary Data 1-Tab 1) was included in the final analysis; two participants from the MICT group and two from the SIT group withdrew from the study due to time constraints, and the data from one participant was excluded from the final analysis as their muscle samples were of poor quality. Samples were also excluded where fibre typing by dot blotting was inconsistent with the proteomic data (see study recruitment flow chart and final group sizes; Supplementary Fig. 1f, overview Supplementary Fig. 1g). The sample size (n) required to reach a significance level of $P < 0.05$ with a power of at least 80% was calculated for the following variables: mitochondrial volume, mitochondrial cristae density, and mitochondrial respiration. Inputted values were based on our prior studies with a lower training volume or duration than the present study. Based on these calculations, a sample size of 8–14 per group was sufficient to reach the level of significance required. Participants were provided with the information to participants' document, and were informed of the study requirements, benefits and risks involved before giving their written informed consent.

**Study design and training.** During the first 2 weeks, participants performed two exercise familiarisation sessions and multiple testing familiarisation sessions. Participants were then allocated into one of the two training groups, based on their age, body mass index (BMI), maximum rate of oxygen consumption ($\dot{V}O_{2max}$), peak aerobic power output ($\dot{W}_{max}$; $\dot{W}.kg^{-1}$) and 20-km time trial (TT) performance ($\dot{W}.kg^{-1}$) (all $P > 0.05$), to match the two groups for baseline endurance characteristics. The participants then underwent another week of testing, had a resting biopsy and then trained 3 to 4 times per week for 8 weeks

before the last muscle biopsy sample and final testing was performed (Fig. 3a).

**Sprint-interval training (SIT).** Eight participants were from this group ($28.3 \pm 5.2$ years; $179 \pm 8$ cm; $77.3 \pm 6.5$ kg; $24.2 \pm 2.6$ BMI; Supplementary Data 1-Tab 1). The SIT group completed 4–8 30-s 'all-out' cycling sprints against a resistance set at 0.075 kg/kg body mass (BM), interspersed with a 4-min recovery period between sprints[69,82]. The training load was increased to 0.080 kg.kg$^{-1}$ BM in week 3, to 0.085 kg.kg$^{-1}$ BM in week 5 and to 0.090 kg.kg$^{-1}$ BM in week 7. During the recovery, participants remained on the bikes and were allowed to either rest or cycle against no resistance. Participants started the training with 4 sprints per session and this increased up to 8 sprints per session in week 7.

**Moderate-intensity continuous training (MICT).** Eight participants were from this group ($26.8 \pm 5.4$ year; $175 \pm 6$ cm; $70.4 \pm 9.6$ kg; $22.9 \pm 2.6$ BMI; Supplementary Data 1-Tab 1). The training intensity for the MICT training group was established as 10% less than the first lactate threshold (~90% of the lactate threshold 1, $LT_1$ - defined as the first increase of $\geq 0.3$ mmol.L$^{-1}$ of lactate from the previous stage during the submaximal test). The training intensity prescription was reassessed during the first training session of weeks 3, 5 and 7 and adjusted accordingly. At the commencement of training, participants completed 60 min per session and this increased up to 120 min per session in week 7.

**Testing procedures.** Participants were required to avoid any vigorous exercise for the 48 h preceding each performance test (72 h for the skeletal muscle biopsy), and to avoid caffeine consumption for at least 8 h prior to each test. Tests were performed at a similar time of the day throughout the study to avoid variations caused by changes in circadian rhythm.

**GXT.** Graded exercise tests (GXTs) were performed on an electronically braked cycle ergometer (Lode Excalibur v2.0, Groningen, The Netherlands) to determine maximal oxygen uptake ($\dot{V}O_{2max}$) and maximal power ($\dot{W}_{max}$). A GXT design of 1-min incremental stages was utilised, aiming to attain a total testing duration of 9–11 min. Breath-by-breath recordings of Oxygen Consumption ($\dot{V}O_2$), Expired Carbon Dioxide ($\dot{V}CO_2$) and minute ventilation ($\dot{V}_E$) were acquired throughout the GXT using a metabolic analyser (Quark Cardiopulmonary Exercise Testing, Cosmed, Italy). The $\dot{W}_{max}$ was determined as the average power of the last minute completed. After completion of the GXT, there was a 5-min recovery followed by a verification bout performed at an intensity equivalent to 90% of $\dot{W}_{max}$ until task failure to confirm the highest measured $\dot{V}O_{2max}$[83].

**Submaximal test.** Once the initial GXT was completed, the ventilatory parameters obtained ($\dot{V}O_2$, $\dot{V}CO_2$, $\dot{V}_E$) were used to estimate the first ventilatory threshold ($VT_1$). Once estimated, the submaximal test

started at 40 W lower than the estimated $VT_1$ and the intensity was increased 10 W every 3 min (stage length) until the $LT_1$ was surpassed and identified. The test was stopped at a blood lactate of 2 mmol.L$^{-1}$ as the $LT_1$ had been reached in all cases. Antecubital venous blood was taken in the last 15 s of each stage and was instantly analysed in duplicate using a blood lactate analyser (YSI 2300 STAT Plus, YSI, USA).

**Physical activity and nutritional controls.** Participants were requested to maintain a normal dietary pattern and physical activity throughout the study. To minimise the variability in muscle metabolism, participants were provided with a standardised dinner (55 kJ/kg of body mass (BM), providing 2.1 g carbohydrate/kg BM, 0.3 fat/kg BM and 0.6 g protein/kg BM) and breakfast (41 kJ/kg BM, providing 1.8 g carbohydrate/kg BM, 0.2 g fat/kg BM and 0.3 g protein/kg BM) to be consumed 15 h and 3 h before the biopsies, respectively.

**Muscle biopsies and single muscle fibre isolation.** A biopsy needle with suction was used to obtain vastus lateralis muscle biopsies under local anaesthesia (1% xylocaine) at pre training and 3 days post training. After being cleaned of excess blood, connective tissue and fat, muscle biopsies were then separated on ice into single fibres. Forty fibres from each biopsy were individually isolated into single fibre segments and then placed in a Laemmli solution (a solubilising buffer)[12]. The Laemmli solution was composed of 4% (v/v) sodium dodecyl sulphate (SDS), a thiol agent of 10% (v/v) ß- mercaptoethanol, 20% (v/v) glycerol, 0.125 M tris-hydroxymethyl-aminomethane (tris)-HCl; and 0.015 % (v/v) bromophenol blue, pH 6.8. Each fibre was diluted with 5 μL of 3x the solubilising buffer 2:1 (v/v) with 10 μL of 1x Tris-Cl (pH 6.8)[12,23]. These fibre segments were then immediately frozen in liquid nitrogen and stored at ⁻80 °C.

**Single muscle fibre analysis—immunoblotting**
**Muscle fibre typing (dot blotting).** To assess muscle fibre types, PVDF membranes were activated in 95% ethanol for 15–60 s and then equilibrated for 2 min in transfer buffer (25 mM Tris, 192 mM glycine, pH 8.3 and 20% methanol). The wet membrane was then placed on a stack of filter paper (one to two pieces soaked in transfer buffer on top of one dry piece). The single-fibre samples in the Laemmli solution were then thawed and vortexed, but not centrifuged, to avoid pelleting and hence loss of any of the skeletal muscle protein[12]. Samples were spotted to a specific part of the membrane in aliquots equating to ~1/8 of a fibre segment (i.e. 1 μL) using a multi-channel pipette. This was repeated twice, once for type I fibre detection and another time for type II fibre detection. After complete absorption of samples, the membrane was placed on top of a dry piece of filter paper to dry for 2 to 5 min before being reactivated in 95% ethanol for 15 to 60 s and equilibrated in transfer buffer for 2 min. After three quick washes in Tris-buffered saline-Tween (TBST), the membrane was blocked in 5% non-fat milk in TBST (blocking buffer) for 5 min at room temperature. Following blocking, the membrane was rinsed with TBST and then incubated in MYH2 dilute 1:200 (#A4.74, Developmental Studies Hybridoma Bank [DSHB]) or MYH7 diluted 1:200 (#A4.840, DSHB) antibody overnight at 4 °C with gentle rocking. On the second day, membranes were washed in TBST and then incubated in secondary antibody IgG (MYH2, #A32723, ThermoFisher Scientific) or IgM (MYH7, # Cat # A-21042 4, ThermoFisher Scientific) diluted 1:20 000 at room temperature for 1 h with rocking. Lastly, membranes were washed in TBST and then exposed to Clarity-enhanced chemiluminescence reagent (BioRad, Hercules, CA, USA), imaged (ChemiDoc MP, BioRad) and analysed for signal density (ImageLab 5.2.1, BioRad) (see example image in Supplementary Fig. 1h). Using images of all the membranes, it was possible to determine the fibre type of each sample (I or IIa) or if no MHC protein was present[84], which would indicate unsuccessful collection of a fibre segment. Samples with signal density

in both type I and IIa membranes were discarded (i.e. a I/IIa hybrid, two different fibre types in the same sample).

**Fibre-type pooling.** The analysis of how the number of detected proteins changed with increasing fibre pooling is shown in Supplementary Fig. 1i. An increasing number of peptide and protein identifications was observed with a greater number of pooled fibres. However, there was little change in the number of detections from 6 to 9 fibre segments and the number of proteins identified plateaued beyond 6 fibres. A fixed fibre pooling of 6 fibre segments was chosen for this study to allow for sufficient protein coverage and to provide reduced heterogeneity with the same number of pooling across all samples. In addition, this was the maximum number of pooled fibres available for some participants.

**Muscle glycogen fibre-type histochemistry analysis.** Muscles were embedded in O.C.T. compound (Tissue-Tek), frozen in liquid nitrogen-cooled isopentane, stored at −80 °C and subsequently cut into 10 μm thick cryosections with a cryostat (Leica) maintained at −20 °C.

Fibre type-specific glycogen depletion was imaged under bright-field using periodic acid-Schiff stain (PAS), similar to the methodology described elsewhere[85]. As such, 10 μm thick muscle cryosections were serially sectioned onto cleaned glass slides and subsequently fixed in 3.7% formaldehyde in 90% ethanol for 60 min at 4 °C. After this fixation, sections were then pre-treated for 5 min with 1% periodic acid (Sigma-Aldrich, Australia) in milliQ-water followed by a washing step for 1 min in tap water and a wash dip for 5 s in milliQ-water. The slides were then applied with 15% diluted Schiff's reagent in PBS (Sigma-Aldrich, Australia) and incubated for 15 min at room temperature. The Schiff's reagent was diluted to 15% based on prior optimisation, as when applied without any dilution the dye intensity saturated the signal. After Schiff's reagent, the sections were washed for 5 s in milliQ-water followed by a 10-min rinse with tap water. Thereafter, sections were washed (3 × 5 min) in PBS (137 mmol.L$^{-1}$ sodium chloride, 3 mmol.L$^{-1}$ potassium chloride, 8 mmol.L$^{-1}$ sodium phosphate dibasic and 3 mmol.L$^{-1}$ potassium phosphate monobasic, pH of 7.4). Slides were then dried before adding ~ 15 μL of PBS per section and mounting with cover slips. The slides were then viewed and imaged in high resolution using an Olympus BX51 fluorescence microscope (Olympus Corporation, Tokyo, Japan) and Cell F software (Olympus). All images were obtained with the x10 objective and captured under bright-field mode.

The second serially sectioned slide underwent immuno-fluorescence staining for MHC expression and was performed with 1:25 primary antibodies against MHCI (BA-F8) and MHCIIa (BF-35) (Developmental Studies Hybridoma Bank, University of Iowa), whereas secondary antibodies were Alexa fluor 350 IgG$_{2b}$ 1:500 (blue) and Alexa fluor 488 IgG$_{2b}$ 1:500 (green) (Invitrogen). Antibody cocktail configurations and immunofluorescence staining procedures were followed according to ref. 86. Once completely dry, the sections were incubated (blocked) with 10% goat serum (product no: 50197Z, ThermoFisher Scientific) for 1 h. The excess goat serum was then removed, and sections were incubated with primary fibre type antibodies over night at 4 °C. The following morning, the primary antibodies were washed off in milliQ-water (3 × 5 min). Once the slides were fully dry, they were incubated with fluoro-conjugated secondary antibodies for 90 min. Slides were washed further with milliQ-water (3 × 5 min) post incubation and dried before adding ~15 μL of PBS per section and mounting with cover slips. The slides were then viewed and imaged in high resolution using an Olympus BX51 fluorescence microscope (Olympus Corporation, Tokyo, Japan) and Cell F software (Olympus). All images were obtained with the x10 objective and analysis of images was performed using ImageJ software (National Institutes of Health, Maryland, USA) (See Fig. 3c for example images).

**Citrate synthase (CS) activity assay.** Citrate Synthase activity was analysed on a 96-well plate by combining 5 μL of a 2 μg μL$^{-1}$ muscle lysate, 40 μL of 3 mM acetyl CoA and 25 μL of 1 mM DTNB to 165 μL of 100 mM Tris buffer (pH 8.3). Following this, 15 μL of 10 mM oxaloacetic acid was added to the plate mixture, protected from the light, and immediately placed in the spectrophotometer at 30 °C (xMark Microplate Spectrophotometer, BioRad Laboratories Pty Ltd, Gladesville, NSW, AUS). Absorbance was recorded at 412 nm every 15 s for 6 min after 30 s of mixture agitation. CS activity was calculated from the steepest part of the curve and reported as mol.kg protein$^{-1}$h$^{-1}$. Two samples from each individual (PRE and POST) were measured in triplicate for Citrate Synthase activity.

**Blood lactate and pH measurements.** Antecubital venous blood samples (~1 mL) were collected pre and post the first exercise training session from a cannula inserted in the antecubital vein for the determination of venous blood [H$^{+}$] and lactate concentrations using a blood-gas analyser (ABL 800 FLEX, Radiometer Copenhagen).

**Transmission electron microscopy.** Skeletal muscle samples were fixed overnight at 4 °C with 0.2 M sodium cacodylate–buffered, 2.5% glutaraldehyde and 2% paraformaldehyde. Fixed samples were rinsed with 0.1 M sodium cacodylate, and postfixed with ferricyanide-reduced osmium tetroxide (1% OsO4, 1.5% K3 [Fe(CN)6], and 0.065 M cacodylate buffer) for 2 h at 4 °C. The postfixed samples, processed by the Monash Ramaciotti Centre for Cryo-Electron Microscopy, were rinsed with distilled water and then stored overnight in 70% ethanol. Dehydration was performed by graduated ethanol series (80%, 90%, 95%, 100% and 100%; 10 min each) and propylene oxide (100% and 100%; 5 min each). Samples were infiltrated with Araldite 502/Embed 812 by graduated concentration series in propylene oxide (25% for 1 h, 33% for 1 h, 50% overnight; 66% for 4 h, 75% for 4 h, 100% overnight; and 100% for 5 h) and then polymerised at 60 °C for 48 h. Embedded samples were sectioned using an Ultracut UCT ultramicrotome (Leica Biosystems) equipped with a 45° diamond knife (Diatome) to cut 75-nm ultrathin sections. The grids were stained at room temperature using 2% aqueous uranyl acetate (5 min) and Reynolds lead citrate (3 min) before routine imaging. All TEM imaging was performed at 80 kV on a Jeol JEM-1400Plus and recorded with a Matataki flash camera and DigitalMicrograph (Version 1.71.38) acquisition software.

Twenty images were randomly captured from a minimum of five fibres in each training group, both before and after training. Eight individuals from each training type were included in this analysis. Mitochondrial volume density was determined using the stereological point counting method as previously described by Broskey et al. [87]. Image quantification was carried out using ImageJ software (NIH, USA), adhering to published guidelines[87]. Image acquisition was conducted in a blinded manner.

**Proteome sample preparation.** Fibres were placed into 5 μL of 2x SDS solubilising buffer (0.125 M Tris·Cl, pH 6.8, 4% SDS, 10% glycerol, 4 M urea, 5% mercaptoethanol, 0.001% bromophenol blue) diluted 2 times (vol/vol) with 1x Tris·Cl, pH 6.8. Fibres were pooled according to fibre type and further lysed with heating at 95 °C for 10 min. The pooled fibres were then sonicated for 20 min (Bioruptor, Diagenode, 20 cycles of 30 s). Reduction and alkylation of disulphides was performed by adding chloroacetamide (40 mM, CAA, Sigma) and further incubated for 30 min at 50 °C. The SDS lysate was acidified with 12% aqueous phosphoric acid at 1:10 for a final concentration of ~ 1.2% phosphoric acid and mixed. This step was essential as the proteins are filtered at this pH. The high percentage of methanol can then precipitate the protein out, which will be retained on top of the S-Trap™ (ProtiFi). Following this, 350 μL of S-Trap™ buffer (90% Methanol (MeOH) and 100 mM triethylammounium bicarbonate (TEAB), C$_7$H$_{17}$NO$_3$) was added to the acidified lysis buffer (final pH 7.1). A colloidal protein

particulate was instantly formed in this step. With the S-Trap™ micro column in a 1.7 mL tube for flow through, the acidified SDS lysate/ MeOH S-Trap™ buffer mixture was added into the micro column. The micro column was then centrifuged at 4000 g for 30 s until all SDS lysate/ S-Trap™ buffer had passed through the S-Trap™ column. Protein was then trapped within the protein-trapping matrix of the spin column. The captured protein was then washed with 350 μL S-Trap™ buffer with centrifugation and washing repeated three times. The spin column was then transferred to a fresh 1.7 mL tube (this aided in preventing contamination of the digestion). For best results, the S-Trap™ micro column was rotated 180 degrees between the centrifugation washes. The S-Trap™ micro column was then moved to a clean 1.7 mL sample tube for the digestion with the protease (trypsin and LysC added at a concentration of 1:50 in 125 μL of 50 mM digestion buffer) into the top of the micro column. The sample was then centrifuged at a low speed of 500 g for 30 s and any solution that passes through is returned to the top of the column (the protein-trapping matrix is highly hydrophilic and will absorb the solution; however, it was important to ensure there was no bubble at the top of the protein trap). The column was then transferred to another fresh tube and incubated at 37 °C overnight (~16 h). For 96-well plates, centrifugation was performed at 1500 g for 2 min. The protease combination was added at a concentration of 1:25 in 125 μL of 50 mM digestion buffer and incubated for 1 h at 47 °C.

Peptides were eluted with 80 μL each of digestion buffer (50 mM TEAB) and then 0.2% aqueous formic acid was added to the S-Trap™ protein trapping matrix centrifuged at 4000 g for 60 s for each elution. Hydrophobic peptides were recovered with an elution of 80 μL 60% (v/v) acetonitrile containing 0.2% formic acid and then centrifuged at 4000 g for 60 s. Elutions were pooled.

Further peptide purification was performed as required with 2x SDB-RPS disc stage tips that were prepared and had sample loaded onto the stage-tip. Peptides were then centrifuged through the column at 1500 g for 3 min. Stage tips were washed with 100 μL of 90% isopropanol (C$_3$H$_8$O) containing 1% TFA and then washed again with 0.2% TFA in 5% acetonitrile and centrifuged for 4 min following each wash. Peptides were then eluted by adding 100 μL of 60% acetonitrile containing 5% ammonium hydroxide and centrifuged for 4 min. Samples were lyophilised down to dryness with the SpeedVac (CentriVap Benchtop Centrifugal Vacuum Concentrator, # 7810038, VWR) and reconstituted for labelling and subsequent MS analysis.

**TMTpro labelling.** TMTpro Label Reagents (ThermoFisher) were equilibrated to room temperature and 20 μL of anhydrous acetonitrile was added to each tube. Dried samples were reconstituted in 0.5 M HEPES buffer, pH 8.5. The desired amount of TMTpro label was added and incubated at 20 °C for 1 h with shaking (1000 rpm). The reaction was then quenched by adding a final concentration of 0.25% of hydroxylamine to the peptide and TMT mixture, and further incubated at 20 °C for 30 min. Samples were lyophilised down to dryness with the SpeedVac and reconstituted in loading buffer. Each sample was fractionated into 16 fractions using basic pH reverse phase C18 liquid chromatography. Peptides were subjected to basic-pH reverse-phase high pressure liquid chromatography (HPLC) fractionation. Labelled peptides were solubilised in buffer A (10 mM ammonium hydroxide) and separated on an Agilent HpH Poroshell120 C18 column (2.7 μm particles, 2.1 mm i.d. and 5 cm in length).

**TMT labelling strategy.** Multiplexing techniques have recently extended from 11 to 16 plex, which not only increases sample throughput but accommodates complex experimental designs such as this training study. This provided the ability to label 16 samples in one run and then merge as one sample through the LC-MS. A 16-plex also allows for an expanded number of treatments, such as replicates and dose-response or time-course measurements can be analysed in the

same experiment with basically no missing values across all samples and extending the statistical power across the entire system[88]. The main study used TMTpro 16-plex version with a different reporter and mass normaliser than earlier TMT versions (i.e. TMT-6 plex). A limitation of tag-based proteomic strategies is ion interference-related ratio distortion resulting from fragmentation and analysis of background ions co-isolated with those of interest. Each sample is differentially labelled, such that when pooled the signal-to-noise values of sample-specific reporter ions represent the relative abundance of each protein. As such, the degree of ion interference by the level of TMT signal detected in channels where a specific protein should be absent can be assessed[89]. Our sampling strategy was adopted to best accommodate for possible reporter ion interferences. Reporter ion interference (RII) targets were classified according to a typical product data sheet for 16-plex TMTpro Label Reagents from ThermoFisher Scientific.

**Liquid chromatography-MS/MS acquisition.** Peptides were loaded onto a 2-cm PepMap trap column (ThermoFisher) and separated using a PepMap 75 µm × 50 cm column (ThermoFisher) with a gradient of 2−30% MeCN containing 0.1% FA over 120 min at 250 nL/min and at 40 °C. The Orbitrap Fusion mass spectrometer was operated with the following parameters: an MS1 scan was acquired from 375−1575 m/z (120,000 resolution, 2e5 AGC, 50 ms injection time) followed by MS2 data-dependent acquisition with collision-induced dissociation (CID) and detection in the ion trap (4e3 AGC, 150 ms injection time, 30 NCE, 1.6 m/z quadrupole isolation width, 0.25 activation Q). To quantify TMTpro reporter ions, a synchronous precursor selection MS3 scan was performed with higher-energy collisional dissociation (HCD) and detection in the orbitrap (120−750 m/z, 1e5 AGC, 250 ms injection time, 60 NCE, 2.5 m/z isolation width). The total cycle time was set to 2.5 s. The acquired raw data was analysed with Proteome Discoverer 2.4 (ThermoFisher) using its implemented SequestHT search engine. Database searching was performed with the following parameters: cysteine carbamidomethylation as well as TMTpro at peptide N-termini and lysine residues was selected as fixed modification, whilst oxidation of methionine and acetylation of protein N-termini were set as variable modifications. Up to 2 missed cleavages were permitted considering a tryptic digestion pattern and the mass tolerance was set to 20 and 10 ppm for precursor and fragment ions, respectively. A false discovery rate (FDR) of 1% was allowed for both protein and peptide identifications. A human protein sequence database downloaded in March 2019 from Uniprot/SwissProt was underlying the searches. The non-normalised protein reporter intensity was exported to Excel and further analysed in R (v3.6.3).

**Bioinformatic analysis of proteomics data**
**Clean up and normalisation.** Only proteins that had a high FDR confidence, were not deemed a contaminant and had greater than one unique peptide, were used for analysis. Proteins having >30% missing data were removed and the remaining missing data was imputed using k nearest neighbour (knn) method from the impute package (v1.76.0) in R (v3.6.3). Normalisation was performed using a combination of trimmed mean of M values (TMM), sample loading (SL) and ComBat (v0.0.4) methods[90–93]. Sample loading adjusts each TMT experiment to an equal signal per channel and TMM uses the average or median signal that is typically applied to use a single multiplicative factor to further adjust the samples to each other. Further to this, ComBat allows users to adjust for batch effects in datasets where the batch covariate is known, using previously described methodology[91].

**Differential expression analysis.** Differential expression analysis was performed using the R package limma (v3.48.3)[51] on the relative protein abundances after first performing the normalisation technique described above. Differential expression values were determined by comparing relative protein abundances. To focus on proteins

specifically associated with fibre type, the differential expression comparisons were first filtered to include proteins differentially expressed between type I and type II fibres before exercise (Benjamini-Hochburg adjusted $P < 0.05$). This established the baseline of the proteins that were significantly different between the two fibre types prior to exercise and allowed for comparisons between our data and the data of refs. 29,30 (Figs. 1 and 2). Following this, comparisons were made to identify any significant effects (Fusion Factor $P < 0.05$) exercise produced on each of the fibre types for each type of exercise (MICT type I Pre vs Post, MICT type II Pre vs Post, SIT type I Pre vs Post, SIT type II Pre vs Post; Supplementary Data 4). Finally, comparisons were completed between fibre types on samples post exercise (MICT Post type I vs type II and SIT Post type I vs type II, Fig. 5a, b, Supplementary Data 5), to distinguish and compare significant (Benjamini-Hochburg adjusted $P < 0.05$) differences between the fibre types after exercise. Mitochondrial proteins were identified by annotation using both the Integrated Mitochondrial Protein Index (IMPI) and the Mito-carta 3.0 database[94].

The heatmap presented in Fig. 1g was completed using the ComplexHeatmap package (v.2.8.0), with hierarchical clustering using the average method. The heatmap presented in Fig. 7c was created using the native R heatmap function and clustered according to the OXPHOS complex subunits as identified in MitoCarta 3.0. For visualisations of differential expression comparisons, volcano plots were presented in Figs. 1f, 5a, b and used the resulting adjusted $P$ values from the limma analysis, which can be found in Supplementary Data 1 and S5. Volcanos from Fig. 4c used the $P$ values from the Fusion Factor calculation found in Supplementary Data 4. Profile Plots presented in Figs. 6d, 7a, b, d and S2b, c, were created by calculating a mean of the group in question (complex or functional group as identified by MitoCarta 3.0) with all proteins identified in that group coloured and a background displayed of all mitochondrial proteins (with an exception for Fig. 7a, which displays a mean of the mitochondrial proteins with a background of all mitochondrial proteins identified). Statistical significance between Pre and Post training values within the profile plots was completed using a paired t-test and the default adjustment for multiple hypothesis testing. Venn diagrams featured in Figs. 1c, 2a, 5c, d, were created using the Venn Diagram (v1.7.1) package.

To better characterise the proteomic changes, gene ontology and enrichment analysis were performed by investigating non-random associations between proteins and over-represented GO. For Fig. 1b, we calculated the average number of proteins in each pooled sample, the proteins identified within the entire dataset of high-confidence proteins and compared them to the entire human genome to determine how many were identified in specific GO CC terms compared to the human genome. Figures 1d and 2b compared the 844 and 314 proteins that were common between the three datasets investigated and the top five GO CC terms for the 844 proteins were visualised, with the top five GO BP terms used for the 314. These analyses were conducted using Database for Annotation, Visualisation and Integrated Discovery (DAVID)[95]. Network plots and WikiPathways are presented in Fig. 5e, f respectively by utilising enrichr (v3.2). The network plot (Fig. 5f) was completed using only statistically significant (Benjamini-Hochburg adjusted $P < 0.05$) results for the comparisons of type I vs II values post both MICT and SIT, whereas the Wikipathways BubblePlot uses all results for the same comparisons. These analyses were completed using the R package ClusterProfiler (v4.0.5).

**Mitochondrial normalisation.** As the mitochondrial protein expression identified a difference in the mitochondrial content between the two fibre types before exercise, we employed a previously reported normalisation strategy[8]. This normalisation strategy removes bias introduced by the differences in mitochondrial content and allows the investigation of these adaptations when this bias was removed.

Mitochondrial proteins identified through IMPI and MitoCarta 3.0 were subset from the rest of the data, and the same normalisation protocol as reported above was applied.

## Reporting summary

Further information on research design is available in the Nature Portfolio Reporting Summary linked to this article.

## Data availability

Source data to interpret, verify and extend this research are provided with this paper. The mass spectrometry proteomics data has been deposited in the ProteomeXchange Consortium via the PRIDE partner repository under accession code PXD036010. Source data are provided with this paper.

## Code availability

The R scripts used for all omics analyses described above are deposited on GitHub and available through https://doi.org/10.5281/zenodo.7227800. There are no restrictions placed on accessibility of this code.

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

## Acknowledgements

We thank Iresha Hanchapola from Monash Proteomics & Metabolomics Facility for proteomic supplies and instrument support and acknowledge Associate Professor Javier Diaz Lara from the University de Navarra for support with dot blotting. This study was supported by grants from the Australian Research Council (Discovery Projects DP140104165 and DP200103542 to D.J.B.), the Australian Physiological Society (Ph.D. grant to J.B.) and the Australian National Health and Medical Research Council (NHMRC Fellowship GNT2009732 to D.A.S.). J.B. was supported by a Victoria University International Postgraduate Research Scholarship.

## Author contributions

D.J.B., J.B., E.G.R. and C.G. conceptualised and designed the study. J.B. carried out the training study and sample collection. E.G.R., J.B. and C.H. performed the experiments. E.G.R. and S.C. performed muscle glycogen fibre-type histochemistry analysis. V.O. and G.R. performed the electron microscopy experiments. N. J. C. performed the statistical and bioinformatic analysis. D.J.B., E.G.R. and N.J.C. wrote the manuscript. E.G.R., J.B., C.H, R.B.S. D.A.S, C.G., N.J.C. and D.J.B. edited and revised the manuscript. All persons designated as authors qualify for authorship, and all those qualifying for authorship are listed. All authors have read and approved the final manuscript.

## Competing interests

The authors declare no competing interests.
