## [Peer Review File · Nature Communications]

Fibre-specific mitochondrial protein abundance is linked to resting and post-training mitochondrial content in the muscle of menREVIEWER COMMENTS

Reviewer #1 (Remarks to the Author):

The Authors provide in-depth analysis of fiber-specific adaptations in human skeletal muscle before and after two different type of training, adopting a TMT/MS-based proteomics workflow and a large sample size. Importantly, the Authors consider the mitochondrial content that characterize type I and Type II fibers introducing a mitochondrial normalisation strategy to select proteins differentially expressed in type I and Type II fibers independently of the different mitochondrial content and associated to training, only. This work is extremely significant in the field of muscle proteome research since results clearly indicate changes in single fiber proteome associated to two different exercise protocols: a mild and prolonged exercise training and sprint interval training and demonstrate that most training-induced changes in mitochondrial protein abundances were linked to the overall increase in mitochondrial content. This strategy can be translated to other paraphysiological and pathological conditions. In addition, the analysis of previous datasets (Murgia et al. 2017 and Deshmukh et al. 2021) provides a robust dataset of the proteomes of fiber type I and II and suggests possible pitfall introduced by different proteomic approaches and of the use of a small sample size. The workflow is described in detail and all methods can be reproduced. The only concern is the absence of hypothesis about Complex IV downregulation after SIT. It could be hypothesized that SIT may induce a feedback inhibition of complex IV by ATP. This “allosteric ATP-inhibition” of phosphorylated and dimeric COX maintains a low and healthy mitochondrial membrane potential (relaxed state), and can prevent the formation of ROS in SIT (doi.org/10.1016/j.mito.2020.10.004). Authors can consider this hypothesis.

Reviewer #2 (Remarks to the Author):

The present study reports proteomic analyses on pooled slow/type I and fast/type II muscle fibers isolated from 16 individuals before and after two types of training (3 to 4 times per week for 8 weeks). The study is focused on mitochondrial changes and, in agreement with previous studies, including three single fibers proteomic analyses, reports that type I fibers have a greater abundance of mitochondrial proteins compared to type II fibers, with some exceptions, and that a moderate-intensity continuous training leads to a global increase in the mitochondrial proteome. The different effect of exercise intensity and the tendency for reduced difference between type I and II fibers post-training was also noted in previous studies. However, the authors also report other data in contrast with previous studies which appear unconvincing (see point 1). Further limitations of this study are discussed below.

Major points

1. The authors report that the mitochondrial content, defined as “mitochondrial protein enrichment (MPE) – a value that identifies the contribution of mitochondrial protein intensities to the overall detectable proteome”, is globally increased by the moderate-intensity continuous training (MICT) protocol (Fig. 6d), but no significant change was detected in specific mitochondrial proteins (Table 4). Indeed, the authors “did not detect any significant changes in protein abundance in either fibre type following either training type” (lines 258 -). This surprising result contrasts with a previous proteomic analysis on the effect of exercise on human pooled type I and II fibers (Deshmukh et al, 2021), in which a significant increase in many TCA cycle and OXPHOS proteins was induced by exercise. Indeed, the authors of the present study previously reported an increase in protein content of OXPHOS subunits assessed in whole-muscle lysates (Granata et al, nc 2021). A large number of biochemical studies, based on enzymatic analyses or western blotting, carried out during the last 50 years (see Holloszy and Booth, 1976 for the early studies), including analyses on single type I and II human muscle fiber pioneered by Oliver Lowry, Bengt Saltin and Jan Henriksson, support the notion that exercise can induce significant changes in the enzyme activity and abundance of mitochondrial proteins leading to functional changes, such as increased respiration. Indeed, the present study reports that citrate synthase (CS) enzyme activity is significantly increased by MICT training (Fig. 6b) but CS protein abundance is unchanged by training, as determined by proteomics (Table 4): this discrepancy is not discussed. It is possible that the apparent lack of significant changes in the abundance of specific mitochondrial protein reported here is due to the different proteomic protocol and bioinformatic analysis procedures. To validate the proteomics results the authors should use an independent approach, such as western blotting for selected mitochondrial proteins, e. g. CS or OXPHOS proteins.

2. The presentation is confused because Results and Discussion are lumped together, thus presentation of data is mixed with interpretation, and it is often difficult to follow a logical sequence. Figures do not match the text in many cases: some figures are not mentioned in the text (e.g. Fig. 4a and Fig. S2d&e), some are not referred in the correct sequence (e.g. Fig. S2 is referred to in page 5 before Fig. S1 in page 7), many panels of different figures are mentioned only in the Methods section, and a general title of each figure is missing due to their heterogeneous composition (e.g. Fig. S2, which is a collage of unrelated data). The supplementary material is presented in 9 “Tables”, some of which are heterogeneous collections of data (see for example Table 1, which contains 7 tabs) and are not presented in a correct sequence (e.g. Table S9 is referred to before Table S2).

Other points

3. A limitation of the present study is that the heterogeneous nature of human fast/type II fibers, most of which are hybrid IIa-IIx fibers containing variable mixtures of MYH2 and MYH1, is not recognized or discussed. Fibers enriched for MYH1, which are a minority in human skeletal muscle and are known to contain higher levels of different muscle proteins (Murgia et al, Skelet Muscle 2021), become invisible using the approach used in the present study. By pooling single muscle

fibers using an antibody that does not distinguish between MYH2 and MYH1 one can increase the yield of identified proteins but completely miss the differences between human fast/type IIa and IIx fiber subpopulations.

4. The authors report that “Our analysis revealed new contractile proteins with fibre-type-specific expression profiles (Table S1-Tab 8). These included MYH3 and MYL4 – two poorly understood members of the myosin family (39)” (Line 168-169). However, i) Tab 8 does not contain only contractile proteins but many proteins with different cellular localization; ii) MYH3 and MYL4 are not “poorly understood members of the myosin family” but well known myosin isoforms present in embryonic skeletal muscle and expressed only in traces in adult muscle; iii) the list contains other proteins which are incorrectly reported: for example, myomesin 2 (MYOM2), a well-known component of the M-band of the sarcomere, is reported to have a location in mitochondria.

5. Important methodological details are missing. For example, neither the legend of Fig. 6 nor the methods specify how many samples from how many individuals, as well as how many EM fields from how many fibers, were used to determine mitochondrial volume density by TEM. Details about citrate synthase activity (how many samples from how many individuals) are also missing.

Reviewer #3 (Remarks to the Author):

The authors studied changes in protein abundance in skeletal muscle fibers after 8 weeks of either moderate-intensity continuous training (MICT) or sprint interval training (SIT). In addition to the usage of two different training models, a major strength is that they studied changes in slow-twitch type 1 and fast-twitch type 2 fibers separately. They focused on mitochondrial proteins and concluded that their major novel finding is that training-induced changes in mitochondrial protein abundance in functional groups are linked to mitochondrial content; in other words, changes reflect mitochondrial quantity rather than quality. The authors also addressed problems with present statistical methods used to detect changes in individual proteins, and they show that detection of functionally relevant training-induced changes, which generally are relatively small (~10-20%), critically depends on the statistical methods.

The study is carefully performed, and methods and results are generally well described. However, there are two exceptions:

1) The increase in mitochondrial volume density (MitoVD) after MICT is a critical factor underlying the conclusion that training-induced changes in mitochondrial proteins are linked to increased mitochondrial content. Thus, the authors must describe how the increase in MitoVD was quantified

(Fig. 6b). The representative images in Fig. 6a indicate that the increase was much larger than the reported up to 4% increase.

2) It is not easy to understand how the p-value adjusted via an empirical Bayes method (limma) was calculated. It is also not clear why the log₂ Fold Difference was set to >0.2 and <-0.2 (Fig. 1f). Please add brief explanations in the Results and Discussion section.

The exercise capacity of the participants was carefully tested before the training periods. However, I cannot find any data on training-induced changes in exercise capacity. From a physiological perspective, it would be very interesting to compare changes in exercise performance with the observed changes in mitochondrial proteins. Please add exercise data if they exist or discuss this shortcoming in the “Limitation of the study” paragraph.

Additional points:

Would it not be better to constantly use “present study” rather than “Reisman et al. 2024” when referring to the present study?

Line 105: While we were...

Fig. 1a, 2nd para: Should it not be “>30% missing data”? The same for line 140 and legend Fig. 4a.

Legend Fig. 3: Add more exact timing of when the post-training biopsy (3a) and blood samples (3b) were taken. Explain that the black circles in Fig. 3b refer to individual values. Explain that decreased glycogen is reflected by reduced staining in Fig. 3c. The red line in Fig. 3d ought to increase at a somewhat lower exercise intensity. Figure 6d shows increases in mitochondrial protein abundance in type 2 fibers after MICT, which must mean that these fibers were activated.

Line 281: Should be “Figure 5c;”

Lines 318-320: It is stated that the changes occurred predominantly in type 1 fibers but in Fig. d, the slope of the black line is steeper for type 2 than for type 1 fibers. Please explain.

The Reference list needs proof reading, e.g. to correct journal titles and formatting of ions.

REVIEWER COMMENTS

Reviewer #1 (Remarks to the Author):

The Authors provide in-depth analysis of fiber-specific adaptations in human skeletal muscle before and after two different type of training, adopting a TMT/MS-based proteomics workflow and a large sample size. Importantly, the Authors consider the mitochondrial content that characterize type I and Type II fibers introducing a mitochondrial normalisation strategy to select proteins differentially expressed in type I and Type II fibers independently of the different mitochondrial content and associated to training, only. This work is extremely significant in the field of muscle proteome research since results clearly indicate changes in single fiber proteome associated to two different exercise protocols: a mild and prolonged exercise training and sprint interval training and demonstrate that most training-induced changes in mitochondrial protein abundances were linked to the overall increase in mitochondrial content. This strategy can be translated to other parapsychological and pathological conditions. In addition, the analysis of previous datasets (Murgia et al. 2017 and Deshmukh et al. 2021) provides a robust dataset of the proteomes of fiber type I and II and suggests possible pitfall introduced by different proteomic approaches and of the use of a small sample size. The workflow is described in detail and all methods can be reproduced. The only concern is the absence of hypothesis about Complex IV downregulation after SIT. It could be hypothesized that SIT may induce a feedback inhibition of complex IV by ATP. This “allosteric ATP-inhibition” of phosphorylated and dimeric COX maintains a low and healthy mitochondrial membrane potential (relaxed state), and can prevent the formation of ROS in SIT (doi.org/10.1016/j.mito.2020.10.004). Authors can consider this hypothesis.

Response:

We appreciate this reviewer’s positive feedback and constructive suggestion to improve our manuscript. After reading the paper suggested by the reviewer, we have added a sentence to the Discussion (Line 438) to further address potential mechanisms that may be influencing COXIV downregulation: ‘It has been hypothesised that complex IV downregulation may be a consequence of allosteric feedback inhibition by ATP¹’.

Reviewer #2 (Remarks to the Author):

The present study reports proteomic analyses on pooled slow/type I and fast/type II muscle fibers isolated from 16 individuals before and after two types of training (3 to 4 times per week for 8 weeks). The study is focused on mitochondrial changes and, in agreement with previous studies, including three single fibers proteomic analyses, reports that type 1 fibers have a greater abundance of mitochondrial proteins compared to type II fibers, with some exceptions, and that a moderate-intensity continuous training leads to a global increase in the mitochondrial proteome. The different effect of exercise intensity and the tendency for reduced difference between type I and II fibers post-training was also noted in previous studies. However, the authors also report other data in contrast with previous studies which appear unconvincing (see point 1). Further limitations of this study are discussed below.

Major points

1. The authors report that the mitochondrial content, defined as “mitochondrial protein enrichment (MPE) – a value that identifies the contribution of mitochondrial protein intensities to the overall detectable proteome”, is globally increased by the moderate-intensity continuous training (MICT) protocol (Fig. 6d), but no significant change was detected in specific mitochondrial proteins (Table 4). Indeed, the authors “did not detect any significant changes in protein abundance in either fibre type following either training type” (lines 258 -). This surprising result contrasts with a previous proteomic analysis on the effect of exercise on human pooled type I and II fibers (Deshmukh et al, 2021), in which a significant increase in many TCA cycle and OXPHOS proteins was induced by

exercise. Indeed, the authors of the present study previously reported an increase in protein content of OXPHOS subunits assessed in whole-muscle lysates (Granata et al, 2021). A large number of biochemical studies, based on enzymatic analyses or western blotting, carried out during the last 50 years (see Holloszy and Booth, 1976 for the early studies), including analyses on single type I and II human muscle fiber pioneered by Oliver Lowry, Bengt Saltin and Jan Henriksson, support the notion that exercise can induce significant changes in the enzyme activity and abundance of mitochondrial proteins leading to functional changes, such as increased respiration. Indeed, the present study reports that citrate synthase (CS) enzyme activity is significantly increased by MICT training (Fig. 6b) but CS protein abundance is unchanged by training, as determined by proteomics (Table 4): this discrepancy is not discussed. It is possible that the apparent lack of significant changes in the abundance of specific mitochondrial protein reported here is due to the different proteomic protocol and bioinformatic analysis procedures. To validate the proteomics results the authors should use an independent approach, such as western blotting for selected mitochondrial proteins, e. g. CS or OXPHOS proteins.

Response:

We thank the reviewer for their comments and recommendations. We agree with the reviewer that the lack of significant changes in individual mitochondrial proteins reported in our manuscript, when compared to Deshmukh et al. 2021, is likely due to different bioinformatic/statistical analyses. It is important to note that our results only contrast with those of Deshmukh et al. 2021 when analysed using a statistical test known as the fusion factor - a test that does not include adjustment for multiple hypothesis testing. As demonstrated within the reanalysis completed by us, and found in Supplementary Table 4 Tab 5, when the original data of Deshmukh et al. were analysed with statistical tests adjusted for multiple hypothesis testing (as recommended by most bioinformaticians), the results in the Deshmukh et al. paper are no longer significant. We add here that the fusion factor method is no longer relied upon by Deshmukh et al., who in more recent papers (i.e. doi: [10.1126/sciadv.adi7548](https://doi.org/10.1126/sciadv.adi7548)), use Benjamini Hochburg adjustment tests with the *limma* pipeline², as completed in this manuscript, to ensure statistical validity. Thus, when adjusted for multiple hypothesis testing, the results of Deshmukh et al. 2021 are consistent with our findings and the small training-induced changes in individual proteins are not identified as significant (as noted by reviewer 3). This probably also explains why changes in CS protein abundance, determined by proteomics in single fibres, did not reach statistical significance but changes in CS activity, determined in whole-muscle samples, did.

While we appreciate the use of independent approaches for validating proteins, unfortunately, due to the in-depth nature of this analysis, we no longer have sufficient remaining muscle to perform western blotting analysis.

2. The presentation is confused because Results and Discussion are lumped together, thus presentation of data is mixed with interpretation, and it is often difficult to follow a logical sequence. Figures do not match the text in many cases: some figures are not mentioned in the text (e.g. Fig. 4a and Fig. S2d&e), some are not referred in the correct sequence (e.g. Fig. S2 is referred to in page 5 before Fig. S1 in page 7), many panels of different figures are mentioned only in the Methods section, and a general title of each figure is missing due to their heterogeneous composition (e.g. Fig. S2, which is a collage of unrelated data). The supplementary material is presented in 9 "Tables", some of which are heterogeneous collections of data (see for example Table 1, which contains 7 tabs) and are not presented in a correct sequence (e.g. Table S9 is referred to before Table S2).

Response:

We thank the reviewer for their comments. We have endeavoured to re- order the figures listed by the reviewer to help in the clarity of the manuscript. We had originally grouped the reference of Figure 4a within the whole of Figure 4, but we have now added Figure 4a specifically to the sections that reference this figure - lines 424, 778,780. We have referenced Figures S2d&e within Line 138.

We have swapped the naming of Figure S2 and Figure S1 within the text and the figure itself, to ensure that they are listed in the correct order. Due to the complex nature of the data, and our view that all data should be open and accessible within the supplementary material for future analysis, we have presented our Supplementary Tables with all the supplementary data that is needed in order to reproduce our analysis for the figures.

Other points

3. A limitation of the present study is that the heterogeneous nature of human fast/type II fibers, most of which are hybrid IIa-IIx fibers containing variable mixtures of MYH2 and MYH1, is not recognized or discussed. Fibers enriched for MYH1, which are a minority in human skeletal muscle and are known to contain higher levels of different muscle proteins (Murgia et al, Skelet Muscle 2021), become invisible using the approach used in the present study. By pooling single muscle fibers using an antibody that does not distinguish between MYH2 and MYH1 one can increase the yield of identified proteins but completely miss the differences between human fast/type IIa and IIx fiber subpopulations.

Response:

We agree with the reviewer that an analysis of type IIx would be extremely valuable within the field; however, this was beyond the scope of this research, as we also wanted to compare with the previous study by Deshmukh et al. 2021. We have added this as a limitation in the appropriate section within the manuscript on Line 465.

4. The authors report that “Our analysis revealed new contractile proteins with fibre-type-specific expression profiles (Table S1-Tab 8). These included MYH3 and MYL4 – two poorly understood members of the myosin family (39)” (Line 168-169). However, i) Tab 8 does not contain only contractile proteins but many proteins with different cellular localization; ii) MYH3 and MYL4 are not “poorly understood members of the myosin family” but well known myosin isoforms present in embryonic skeletal muscle and expressed only in traces in adult muscle; iii) the list contains other proteins which are incorrectly reported: for example, myomesin 2 (MYOM2), a well-known component of the M-band of the sarcomere, is reported to have a location in mitochondria.

Response:

Point i) We agree that Tab 8 contains a variety of different proteins with different cellular localizations. As noted in the header for this tab, the data here are “New proteins identified in the present study as having fibre-type-specific expression profiles.” In general, our approach is to be transparent and provide all of our data for scrutiny by the scientific community. Nonetheless, in the text, rather than discuss all of the proteins, we wanted to highlight the contractile proteins as they are of note to the exercise science community.

Point ii) We agree with the reviewer that the language used to describe the members of the myosin family was not specific enough. We have changed line 170-172 to specify – “*Our analysis revealed new contractile proteins with fibre-type-specific expression profiles (Table S1-Tab 8). These included MYH3 and MYL4 – two poorly understood members of the myosin family in the context of exercise.*”

For point iii) This information from the MitoCarta 3.0 database highlights the main location of the human ortholog in the Protein Atlas based on antibody-based high-resolution microscopy in multiple cell lines, along with an annotation of reliability, as of 2020. This information has been carried over from MitoCarta 3.0 and can be identified within the original database downloaded from the broadinstitute.org page.

5. Important methodological details are missing. For example, neither the legend of Fig. 6 nor the methods specify how many samples from how many individuals, as well as how many EM fields from how many fibers, were used to determine mitochondrial volume density by TEM. Details about

citrate synthase activity (how many samples from how many individuals) are also missing.

Response:

Thank you to the reviewer for this comment. We have added additional methodological details to both the legend of Fig. 6 as well as within the methods.

Line 663 reads: 'Twenty images were randomly captured from a minimum of five fibers in each training group, both before and after training. Eight individuals from each training type were included in this analysis. Mitochondrial volume density was determined using the stereological point counting method as previously described by Broskey et al. ³. Image quantification was carried out using ImageJ software (NIH, USA), adhering to guidelines in ³. Image acquisition was conducted in a blinded manner.'

Line 1349 reads: Twenty images were randomly obtained from at least five fibres of eight individuals of each training group and quantified Pre and Post.

Line 638 & 1353 reads: Two samples from each individual (PRE and POST) were measured in triplicate for Citrate Synthase activity.

Reviewer #3 (Remarks to the Author):

The authors studied changes in protein abundance in skeletal muscle fibers after 8 weeks of either moderate-intensity continuous training (MICT) or sprint interval training (SIT). In addition to the usage of two different training models, a major strength is that they studied changes in slow-twitch type 1 and fast-twitch type 2 fibers separately. They focused on mitochondrial proteins and concluded that their major novel finding is that training-induced changes in mitochondrial protein abundance in functional groups are linked to mitochondrial content; in other words, changes reflect mitochondrial quantity rather than quality. The authors also addressed problems with present statistical methods used to detect changes in individual proteins, and they show that detection of functionally relevant training-induced changes, which generally are relatively small (~10-20%), critically depends on the statistical methods.

The study is carefully performed, and methods and results are generally well described. However, there are two exceptions:

1) The increase in mitochondrial volume density (MitoVD) after MICT is a critical factor underlying the conclusion that training-induced changes in mitochondrial proteins are linked to increased mitochondrial content. Thus, the authors must describe how the increase in MitoVD was quantified (Fig. 6b). The representative images in Fig. 6a indicate that the increase was much larger than the reported up to 4% increase.

Response:

We appreciate this comment from the Reviewer and have added the following at Line 664 to explain the quantification of MitoVD:

Line 663 reads: *Twenty images were randomly captured from a minimum of five fibers in each training group, both before and after training. Eight individuals from each training type were included in this analysis. Mitochondrial volume density was determined using the stereological point counting method as previously described by Broskey et al. ³. Image quantification was carried out using ImageJ software (NIH, USA), adhering to guidelines in ³. Image acquisition was conducted in a blinded manner.*

It should also be noted that the MitoVD data shown in Figure 6b indicate the absolute change (Δ) in mitochondrial volume density (MitoVD – the proportion of the muscle volume occupied by mitochondria). For clarity, we have provided a more detailed description in the figure legend on line 1347.

2) It is not easy to understand how the p-value adjusted via an empirical Bayes method (limma) was calculated. It is also not clear why the log₂ Fold Difference was set to >0.2 and <-0.2 (Fig. 1f). Please add brief explanations in the Results and Discussion section.

Response:

We thank the reviewer for their comments. As requested, we have added a brief description/explanation to the figure legend and also the results section:

Fig 1f legend: *“Points were coloured if they had an adjusted P-value < 0.05 and coloured red with a Log₂ Fold Difference of > 0.2 (higher in Type II fibres) or blue with a Log₂ Fold Difference of < -0.2 (higher in Type I fibres). Points were labelled if they were identified as differentially expressed uniquely within our study (compared with the differentially expressed proteins identified in Deshmukh et al. 2021 and Murgia et al. 2017).”*

Line 163: *‘Proteins were coloured if they had an adjusted P-value < 0.05 and a Log₂ Fold Difference of > 0.2 (red, higher in Type II fibres) or < -0.2 (blue, higher in Type I fibres) (Fig. 1f); for full listing of significant proteins please see Table S1- Tab 8, with labelled points indicating proteins that were uniquely differentially expressed within our study (compared with Deshmukh et al. 2021 and Murgia et al. 2017).’*

We also understand that there was a lack of the usage of the term ‘Benjamini-Hochburg’ adjusted method within the manuscript. We have added ‘Benjamini-Hochburg’ to lines 162, 773, 781, 810, 1333, within the results and discussion to ensure there is clarity about the method used.

The exercise capacity of the participants was carefully tested before the training periods. However, I cannot find any data on training-induced changes in exercise capacity. From a physiological perspective, it would be very interesting to compare changes in exercise performance with the observed changes in mitochondrial proteins. Please add exercise data if they exist or discuss this shortcoming in the “Limitation of the study” paragraph.

Response:

To address the reviewers’ concerns, we have added some additional data for the average of the participants and their performance from PRE to POST training for the two training groups (MICT and SIT). This data is now found in Supplementary Table 1 Tab 1.

Additional points:

Would it not be better to constantly use “present study” rather than “Reisman et al. 2024” when referring to the present study?

Response:

We agree with the reviewer and have changed all references to the manuscript as, ‘present study’ (Lines 1218, 1292, 1299).

Line 105: While we were...

Response:

This line has been changed in accordance with the reviewer’s suggestion on line 105.

Fig. 1a, 2nd para: Should it not be “>30% missing data”? The same for line 140 and legend Fig. 4a.

Thank you to the reviewer for highlighting this; we have corrected the instances on line 140 and line 1295 to read, ‘if they contained more than 30% missing data’.

Legend Fig. 3: Add more exact timing of when the post-training biopsy (3a) and blood samples (3b) were taken. Explain that the black circles in Fig. 3b refer to individual values. Explain that decreased glycogen is reflected by reduced staining in Fig. 3c. The red line in Fig. 3d ought to increase at a somewhat lower exercise intensity. Figure 6d shows increases in mitochondrial protein abundance in type 2 fibers after MICT, which must mean that these fibers were activated.

We have added extra information to the legend for Fig 3a, to highlight the exact timing when the biopsies were taken. This now reads, 'Biopsies were taken at rest before the first session and 72 h post the final exercise session.' on Line 1262. We have also added when the blood samples were taken, which now reads: 'Antecubital venous blood samples were drawn at rest and immediately post the exercise session.' on Line 1273. We have added to 3b that, 'Individual points signify individual participant values' on Line 1273 and to Fig 3c that a 'decrease in staining saturation reflects lower glycogen concentration.' on Line 1281. Figure 3d was based on a previously published review from Egan and Zierath in *Cell Metabolism*, 2013⁴, but we agree with your suggestion and the red line now starts at a slightly lower intensity.

Line 281: Should be "Figure 5c;"

Response:

Line 283 has been changed to Figure 5c.

Lines 318-320: It is stated that the changes occurred predominantly in type 1 fibers but in Fig. d, the slope of the black line is steeper for type 2 than for type 1 fibers. Please explain.

Response:

We thank the reviewer for their comments and have assumed that the figure mentioned is Figure 6d. MICT Type II from PRE to POST is more significant and we have changed lines 320 to reflect this which now read, "Changes in the mean abundance of known mitochondrial proteins following MICT occurred in both the type I and type II fibres (Figure 6d); this suggests that there is recruitment of type II fibres along with type I fibres, during long-duration, moderate-intensity exercise. There was a significant increase in known mitochondrial protein abundance in fibre type I following SIT (Figure 6d), with a small decrease observed. "

The Reference list needs proof reading, e.g. to correct journal titles and formatting of ions.

We have endeavoured to check the reference list as closely as possible, with limitations existing due to the referencing outline provided for Nature journals within EndNote. We are able to change references at the request of the Editor to fit the format if necessary.

References used in our response to the reviewers:

1. Kadenbach B. Complex IV – The regulatory center of mitochondrial oxidative phosphorylation. *Mitochondrion* **58**, 296-302 (2021).
2. Ritchie ME, *et al.* limma powers differential expression analyses for RNA-sequencing and microarray studies. *Nucleic acids research* **43**, e47 (2015).
3. Broskey NT, Daraspe J, Humbel BM, Amati F. Skeletal muscle mitochondrial and lipid droplet content assessed with standardized grid sizes for stereology. *Journal of Applied Physiology* **115**, 765-770 (2013).

4. Egan B, Zierath JR. Exercise metabolism and the molecular regulation of skeletal muscle adaptation. *Cell Metab* **17**, 162-184 (2013).

REVIEWER COMMENTS

Reviewer #2 (Remarks to the Author):

Concerning point 1. I am still surprised that the authors did not detect any significant changes in protein abundance in either fiber type following either training type. In their rebuttal the authors discuss at length the difference in the bioinformatic analysis with Deshmukh et al, but the point remains, as I stressed in my original comment, that these results contrast with dozens of biochemical studies on whole muscle and single fibers during the last 50 years showing that different mitochondrial proteins do increase after protocols of endurance exercise comparable to the MICT training used here. The present work shows that the expected increase is only seen with the whole mitochondrial proteome or mitochondrial functional groups, such as TCA cycle proteins, but surprisingly not OXPHOS. However, specific enzymes of the TCA cycle, such as citrate synthase do not show significant change, although the authors show that CS enzymatic activity of whole muscle samples is significantly increased, in agreement with numerous previous studies. The authors do not discuss the reason for this discrepancy, whether this may be due to the variability of proteomic data or too stringent statistical approaches or other reasons.

Both in the abstract and in the text the authors stress the point that “Most training-induced changes in different mitochondrial functional groups, in both fibre types, were stoichiometrically linked to changes in markers of mitochondrial content.” This conclusion, which is the opposite of the conclusion in their previous paper (Granata et al. High-intensity training induces non-stoichiometric changes in the mitochondrial proteome ... nc 2021) is based on the very few changes detected in mitochondrial groups, such as TCA cycle and ribosomal proteins. However, the authors also point out that changes in fatty acid oxidation in type I fibers after MICT and in complex IV after SIT are not stoichiometrically linked to changes in global mitochondrial content. Most importantly, no training-dependent change is observed in other major mitochondrial groups, such as OXPHOS, and in any single mitochondrial protein, thus one cannot draw any reasonable conclusion about stoichiometry or non-stoichiometry.

Concerning our suggestion to perform western blotting on selected mitochondrial protein to validate the proteomic results, the authors respond they no longer have sufficient remaining muscle to perform this analysis. This is unfortunate, they should have prepared additional fiber pools to be used as a reserve. WB analyses have been successfully performed on single muscle fibers (Murphy and Lamb, J Physiol 2013). Using additional fiber pools for WB analyses, Deshmukh et al. (2021) have shown that proteins of the OXPHOS complexes are increased by training in both fiber types, thus supporting proteomic data. A similar control should have been done in the present study, given the surprising finding (Fig. 7b) that no significant change in any OXPHOS complex is detectable by proteomics after MICT training. The lack of material to perform further biochemical

analyses of this dataset should not be considered as a valid justification to not perform further experiments requested by reviewers, but rather as a serious fault in the study design.

Concerning point 3: The fact that type IIX fibers were not considered, thus the type II fibers in this study are a mixture of type IIA, type IIX and hybrid IIA/x fibers, is a serious limitation, which might contribute to increase the variability of proteomic data in the type II fiber population.

Concerning point 4: ii) The new version reads "...These included MYH3 and MYL4 – two poorly understood members of the myosin family in the context of exercise." The authors should specify "... MYH3 and MYL4, which code for myosin heavy and light chains present in embryonic skeletal muscle, whose presence in adult muscle is usually interpreted as a sign of muscle regeneration". iii) I am surprised that the authors insist that MYOM2 (myomesin 2) is in the MitoCharta 3.0 database. This protein is NOT in MitoCharta 3.0 and is a well-known component of the M-band of the sarcomere.

Reviewer #3 (Remarks to the Author):

The authors have adequately addressed all points raised in my initial review with one exception:

Some data of training-induced changes in performance are now presented in a Supplementary table, and it seems like maximal oxygen uptake and maximum power during the exercise test increased to a similar extent with both training types. I understand that proteomics is the focus of the study. However, from a physiological perspective it would still be interesting to relate the observed fiber type-dependent changes in protein expressions to changes in performance. Thus, I recommend some mentioning of the observed changes in performance in the Results, and a very brief discussion of their relation to the proteomics results in the Discussion.

REVIEWER COMMENTS

Reviewer #2 (Remarks to the Author):

Point 1. I am still surprised that the authors did not detect any significant changes in protein abundance in either fiber type following either training type. In their rebuttal the authors discuss at length the difference in the bioinformatic analysis with Deshmukh et al, but the point remains, as I stressed in my original comment, that these results contrast with dozens of biochemical studies on whole muscle and single fibers during the last 50 years showing that different mitochondrial proteins do increase after protocols of endurance exercise comparable to the MICT training used here. The present work shows that the expected increase is only seen with the whole mitochondrial proteome or mitochondrial functional groups, such as TCA cycle proteins, but surprisingly not OXPHOS. However, specific enzymes of the TCA cycle, such as citrate synthase do not show significant change, although the authors show that CS enzymatic activity of whole muscle samples is significantly increased, in agreement with numerous previous studies. The authors do not discuss the reason for this discrepancy, whether this may be due to the variability of proteomic data or too stringent statistical approaches or other reasons.

Authors' response to point 1: We agree with the reviewer, and we were also initially surprised by this result. However, there are a couple of relevant points worth noting:

1. The first is that our results are in strong agreement with the only published, comparable study – that of Deshmukh et al.¹ (when we reanalysed their data with the same imputation and empirical Bayes methods as our study; Table S4). While some may contend that this statistical approach is too stringent, most statisticians/bioinformaticians support the need to control for false positives using an adjusted p-value when analysing large data sets². Nonetheless, for readers who consider our statistical approach too stringent, we have provided all of the individual data, all of the individual fold changes, and all of the less-stringent paired t-test values (Table S4); readers are, therefore, able to interpret our data on the basis of less stringent statistical controls, should they wish. Regarding this, in our study, there are hundreds of individual proteins (including many OXPHOS components and other mitochondrial proteins) that have a paired t-test value for the change with training that is less than 0.05.
2. We also agree that western blot studies on single fibers have often reported that different mitochondrial proteins do increase after protocols of endurance exercise comparable to the MICT training used here (as we highlight in the introduction). However, the point about appropriate statistical controls when performing multiple comparisons is also relevant when interpreting the findings of single-fiber studies. When we reanalyse the findings of published single-fiber studies, correcting for multiple comparisons, many of the reported changes for individual proteins are no longer significant. Thus, the question of appropriate statistical controls when analysing multiple proteins, in multiple fibre types, has often been overlooked but is also important in the context of western blot studies. Small fold changes in individual proteins may be significant when analysed with a paired t-test (or similar) but are often no longer significant when analysed in the context of changes in many other proteins.

While we appreciate the opportunity to elaborate on this issue in our response to the reviewer's comments, we agree with the reviewer that a more detailed discussion of this important point is warranted. We have rewritten the paragraph discussing this data in the results (lines 263-283). In addition, we have also discussed the need for further research to be conducted on an increased number of samples in the limitations paragraph on line 473 - "To better account for the inherent inter-individual/inter-fibre variability of human muscle biopsy samples, and the individual variability in the response to training, studies with even larger sample sizes should also be considered."

Point 2. Both in the abstract and in the text the authors stress the point that “Most training-induced changes in different mitochondrial functional groups, in both fibre types, were stoichiometrically linked to changes in markers of mitochondrial content.” This conclusion, which is the opposite of the conclusion in their previous paper (Granata et al. High-intensity training induces non-stoichiometric changes in the mitochondrial proteome ... nc 2021) is based on the very few changes detected in mitochondrial groups, such as TCA cycle and ribosomal proteins. However, the authors also point out that changes in fatty acid oxidation in type I fibers after MICT and in complex IV after SIT are not stoichiometrically linked to changes in global mitochondrial content. Most importantly, no training-dependent change is observed in other major mitochondrial groups, such as OXPHOS, and in any single mitochondrial protein, thus one cannot draw any reasonable conclusion about stoichiometry or non-stoichiometry.

Authors’ response to point 2. Thanks for this suggestion. We agree that there is not sufficient evidence to make robust conclusions about stoichiometry (or non-stoichiometry) and have removed all mentions of stoichiometry from both the abstract and the text.

Point 3. Concerning our suggestion to perform western blotting on selected mitochondrial protein to validate the proteomic results, the authors respond they no longer have sufficient remaining muscle to perform this analysis. This is unfortunate, they should have prepared additional fiber pools to be used as a reserve. WB analyses have been successfully performed on single muscle fibers (Murphy and Lamb, J Physiol 2013). Using additional fiber pools for WB analyses, Deshmukh et al. (2021) have shown that proteins of the OXPHOS complexes are increased by training in both fiber types, thus supporting proteomic data. A similar control should have been done in the present study, given the surprising finding (Fig. 7b) that no significant change in any OXPHOS complex is detectable by proteomics after MICT training. The lack of material to perform further biochemical analyses of this dataset should not be considered as a valid justification to not perform further experiments requested by reviewers, but rather as a serious fault in the study design.

Authors’ response to point 3: We acknowledge that previous research has used western blot results to complement quantitative proteomics results. However, proteomics is a direct observation of the peptide, and, with improvements in sensitivity and reliability of mass-spectrometry-based proteomics, many scientists consider western blots, which rely on single antibodies that may have limited characterisation of their affinity to the antigen and/or its epitope, no longer necessary to validate proteomics^{3, 4, 5, 6}. The quantification of proteins through western blotting relies on a singular signal: the intensity observed in a band on the blot. This signal may either be specific, accurately representing the targeted protein, or non-specific. Furthermore, as highlighted in our response to point 1, an often-overlooked issue when ‘validating’ protein changes with western blots is that correcting for multiple comparisons is often not performed. This probably explains why the western blot results of Deshmukh et al. (2021) are consistent with their proteomics results - when analysed with non-adjusted statistical tests – but not when their proteomics results are corrected for multiple comparisons. Thus, while we advocate for supporting findings with alternative techniques, there are many reasons to question whether western blots are necessary and appropriate to ‘validate’ proteomics; we add that, regardless, this was not possible for our experiment due to the limited amount of human muscle tissue obtained via the needle biopsy technique (which had to be used for many types of analyses).

Point 4: The fact that type IIx fibers were not considered, thus the type II fibers in this study are a mixture of type IIa, type IIx and hybrid IIa/x fibers, is a serious limitation, which might contribute to increase the variability of proteomic data in the type II fiber population.

Authors' response to Point 4: While we agree that considering type IIx fibres would have been interesting, we don't agree that it is a serious limitation. As noted by leaders in the field (e.g., Schiaffino⁷, Schiaffino & Reggiani⁸, and Harridge et al.⁹), human skeletal muscles contain mostly type I and IIa fibres, with IIx fibres being a relatively minor component in most individuals. More recent research, which included individual values, reported that most of their young participants had no type IIx or IIa/x fibres¹⁰. Thus, it is very unlikely that we would have been able to identify and pool sufficient numbers of type IIx or IIa/x fibres in an adequately powered number of our participants for our analyses. We also note that our protocol was based on Deshmukh et al. 2021 (also published in this journal), in which they performed proteomics on pooled slow- (type I) and fast-twitch (type IIa and IIx) muscle fibers.

Nonetheless, acknowledging the important observation of the reviewer, we have added the following to our limitations section on line 469: "Additionally, and similar to previous research¹, the type II fibre pools for some participants may have included a small proportion of type IIx and/or type IIa/x fibres^{7, 11}. Future research in this area should also consider analysing type IIx fibres, which may help explain some of the variability of the proteomic data in the type II fibre pools reported by us and others."

Point 5: ii) The new version reads "...These included MYH3 and MYL4 – two poorly understood members of the myosin family in the context of exercise." The authors should specify "... MYH3 and MYL4, which code for myosin heavy and light chains present in embryonic skeletal muscle, whose presence in adult muscle is usually interpreted as a sign of muscle regeneration". iii) I am surprised that the authors insist that MYOM2 (myomesin 2) is in the MitoCharta 3.0 database. This protein is NOT in MitoCharta 3.0 and is a well-known component of the M-band of the sarcomere.

Authors' response to Point 5: We thank the reviewer for their comments and have changed Line 175 to read, "These included MYH3 and MYL4, which code for myosin heavy and light chains present in embryonic skeletal muscle; their presence in adult muscle is usually interpreted as a sign of muscle regeneration", as per the reviewer's request. We have also removed MYOM2 from our MitoCarta database and it is no longer listed as a mitochondrial protein.

Reviewer #3 (Remarks to the Author)

The authors have adequately addressed all points raised in my initial review with one exception: Some data of training-induced changes in performance are now presented in a Supplementary table, and it seems like maximal oxygen uptake and maximum power during the exercise test increased to a similar extent with both training types. I understand that proteomics is the focus of the study. However, from a physiological perspective it would still be interesting to relate the observed fiber type-dependent changes in protein expressions to changes in performance. Thus, I recommend some mentioning of the observed changes in performance in the Results, and a very brief discussion of their relation to the proteomics results in the Discussion.

Authors' response. We really appreciate the reviewer's perspective and have debated this at length amongst the authors. Ultimately, we are concerned that it is too speculative to make robust conclusions about the contribution of changes to individual proteins, in different fiber pools, to whole-body performance. In this regard, we have added the following to the limitations section (line 475): "*It was not possible for us to determine if the fibre-specific adaptations to the two different types of exercise training had functional consequences, but it is unlikely that small changes in individual proteins are able to explain the observed improvements in exercise performance.*"

References used in this response

1. Deshmukh AS, *et al.* Deep muscle-proteomic analysis of freeze-dried human muscle biopsies reveals fiber type-specific adaptations to exercise training. *Nature Communications* **12**, 304 (2021).
2. Kelter R. Analysis of type I and II error rates of Bayesian and frequentist parametric and nonparametric two-sample hypothesis tests under preliminary assessment of normality. *Computational Statistics* **36**, 1263-1288 (2021).
3. Mehta D, Ahkami AH, Walley J, Xu S-L, Uhrig RG. The incongruity of validating quantitative proteomics using western blots. *Nature Plants* **8**, 1320-1321 (2022).
4. Abbatiello SE, *et al.* Large-Scale Interlaboratory Study to Develop, Analytically Validate and Apply Highly Multiplexed, Quantitative Peptide Assays to Measure Cancer-Relevant Proteins in Plasma *. *Molecular & Cellular Proteomics* **14**, 2357-2374 (2015).
5. Collins BC, *et al.* Multi-laboratory assessment of reproducibility, qualitative and quantitative performance of SWATH-mass spectrometry. *Nature Communications* **8**, 291 (2017).
6. Aebersold R, Burlingame AL, Bradshaw RA. Western blots versus selected reaction monitoring assays: time to turn the tables? *Mol Cell Proteomics* **12**, 2381-2382 (2013).
7. Schiaffino S. Fibre types in skeletal muscle: a personal account. *Acta Physiologica* **199**, 451–463 (2010).
8. Schiaffino SaR, C. Fiber Types in Mammalian Skeletal Muscles. *Physiol Rev* **91**, 1447–1531 (2011).
9. Harridge SD, *et al.* Whole-muscle and single-fibre contractile properties and myosin heavy chain isoforms in humans. *Pflugers Archiv : European journal of physiology* **432**, 913-920 (1996).
10. St-Jean-Pelletier F, *et al.* The impact of ageing, physical activity, and pre-frailty on skeletal muscle phenotype, mitochondrial content, and intramyocellular lipids in men. *J Cachexia Sarcopenia Muscle* **8**, 213-228 (2017).
11. Bottinelli R, Reggiani C. Human skeletal muscle fibres: molecular and functional diversity. *Prog Biophys Mol Biol* **73**, 195-262 (2000).